# SAMPLE-EFFICIENT REINFORCEMENT LEARNING BY WARM-STARTING WITH LLMS

## ABSTRACT

We investigate the usage of Large Language Models (LLMs) in collecting high-quality data to warm-start Reinforcement Learning (RL) algorithms for learning in Markov Decision Processes (MDPs). Specifically, we leverage the in-context decision-making capability of LLMs, to generate an "offline" dataset that sufficiently covers state-actions visited by some good policy, then use an off-the-shelf RL algorithm to further explore the environment and fine-tune its policy, in a black-box manner. Our algorithm, LORO[1], can both converge to an optimal policy and have a high sample efficiency thanks to the good data coverage collected by the LLM. On multiple OpenAI Gym environments, such as CartPole and Pendulum, given the same environment interaction budget, we empirically demonstrate that LORO outperforms baseline algorithms such as pure LLM-based policies, pure RL, and a naive combination of the two.

## 1 INTRODUCTION

The standard protocol in online RL has many applications, from playing games Silver et al. (2017) to robotic control Kober et al. (2013). While having impressive empirical performance and enjoying the theoretical guarantee on returning the optimal policy under some assumptions (Ramaswamy & Hüllermeier, 2021; Agarwal et al., 2019; Bertsekas, 2007), a key problem of this approach is its sample inefficiency, which limits its applications in practice Yu (2018). Thus, most impressive successes in online RL have been restricted to settings where many samples can be obtained by interacting with the environment (such as games or environments with high-quality simulations).

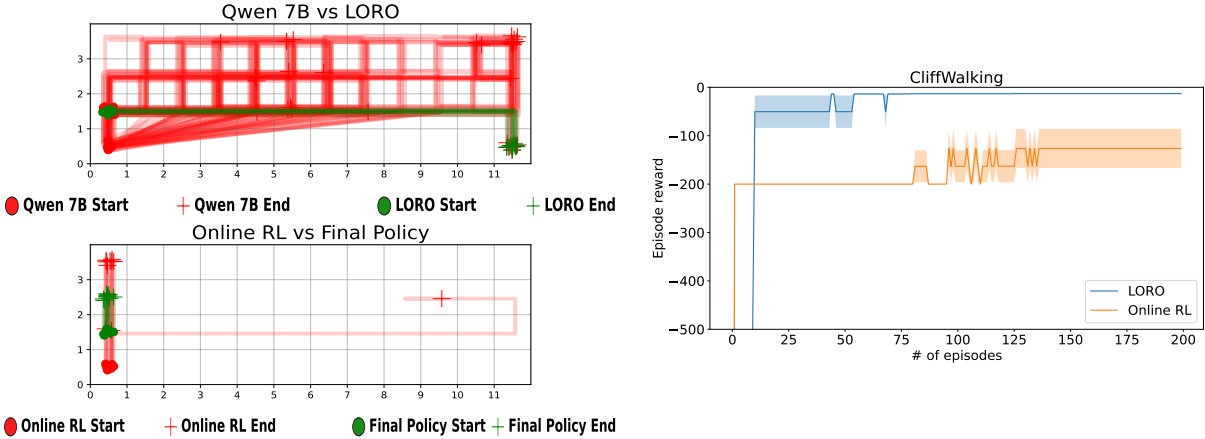

Figure 1: The CliffWalking "offline" dataset (red) collected from a LLM covers the optimal policy much more than the ones collected from a vanilla Online RL or Uniformly Random policy, making the cumulative reward (-9213) three times larger than the Online RL baseline (-37151).

To address this, (Lange et al., 2012; Ernst et al., 2005; Riedmiller, 2005; Levine et al., 2020) proposed the offline RL setting, where the algorithm does not directly interact with the environment as in online RL, but is trained on

---

[1]The code of our experiments can be viewed at `https://anonymous.4open.science/r/LlamaGym-551D`

a large dataset of experience collected from some other sources (e.g., by expert demonstration). While the sample efficiency problem is mitigated due to the large training dataset, these offline RL methods suffer from the *distribution shift* problem, where the state distribution from the offline data differs significantly from the one induced by online interactions Wang et al. (2021).

A popular approach to address the distribution shift problem is by aggregating both the offline and online data (Xie et al., 2021; Song et al., 2022; Zhang & Zanette, 2023). This offline-to-online approach greatly reduces the sample requirement of RL by reducing unnecessary exploration with the offline dataset while also mitigating the distribution shift problem through online interactions. Under some assumptions, Song et al. (2022) provides a cumulative regret and sample complexity guarantee for the offline-to-online setting. They show that, if the offline data distribution covers some high-quality policies' trajectories, their offline-to-online algorithm is both sample-efficient and competitive with the high-quality policies covered by the offline data.

Even then, our goal is to further improve the sample efficiency, perhaps by leveraging extra information from the problem description and world knowledge (e.g., avoid obstacles, find the key to open the door, etc.). Recently, LLM has shown a remarkable ability for memorizing world knowledge and reasoning, even in hard sequential decision problems such as robot manipulation (Ahn et al., 2022; Huang et al., 2022; Liang et al., 2023). Even though LLM needs knowledge about the environments for prompt design, this requirement can still be satisfied for many real-world applications, especially when the problem descriptions and documentations are available publicly on the internet. Thus, we raise the question:

*Can LLM produce a good coverage dataset to boost the data efficiency in RL via warm-starting?*

In this paper, we answer this question positively. Under Assumption 1, where the policy suggested by the LLM has sufficient coverage of an optimal policy, our algorithm, LLM Offline, RL Online (LORO), enjoys both small Cumulative regret suboptimality and Sample complexity. We verify this coverage assumption in Figure 1 and Section 5.2. To the best of our knowledge, we are the first to suggest warm-starting RL with LLM's collected data and connect LLM with offline-to-online RL, drawing the similarity between the distribution shift problem in Offline RL versus the useful-but-suboptimal policy extracted from LLM, and suggest that the offline-to-online RL approaches can be applied here. We verified this suggestion by showing that our LORO algorithm achieves the best overall performance in six environments in Table 1. We also demonstrate the flexibility of our LORO algorithm by showing a similar performance boost when we use a different RL algorithm, such as AWAC Nair et al. (2020) (Section 5.5). Empirically, we demonstrate the effectiveness of our algorithm in four out of six OpenAI Gym environments Towers et al. (2024), improving the cumulative reward up to four times other baselines such as pure RL, pure LLM-based policies, and a naive combination of the two.

## 2 RELATED WORK

**Offline-to-online RL.** Nair et al. (2020) showed that a naive combination of offline pre-training and online fine-tuning does not usually help and often worsens the performance, a large part due to excess conservatism when utilizing the offline data (Fujimoto et al., 2019; Kumar et al., 2019). In addition, in the simple Bandits setting, Sentenac et al. (2025) suggests that the degree of conservativeness, or the amount of exploration, should be different conditioned on the amount of offline versus online data. Different than many previous offline-to-online works, our paper does not focus on efficient use of an offline dataset or addressing the distribution shift problem. We propose that, when there is no offline data available, we can use an LLM to collect a small offline dataset, which can be useful for warm-start learning. We also pointed out the sub-optimality problem with the LLM policy and suggested that prior approaches to address the distribution shift problems listed above can be applied to our approach as well.

**Coverage in offline-to-online RL.** Under some assumptions, Song et al. (2022) provides the cumulative regret suboptimality and sample complexity guarantees conditioned on the Transfer Coefficient that describes the coverage of the offline dataset with an optimal policy. Foster et al. (2025)'s analysis also shows that the data efficiency of any algorithm that run in polynomial time and returns an $\varepsilon$-optimal policy with high probability is lower bounded by a coverage notion that is closely related to Song et al. (2022)'s definition.

**Warm-starting RL.** Schmitt et al. (2018) propose to kick-start Deep RL with a teacher policy by adding an extra objective to encourage the learner to behave similarly to the teacher, with a diminishing weight to allow the student to

eventually surpass the teacher. One limitation is that Schmitt et al. (2018) assumes the teacher policy is high-performing enough to be distilled, meaning its application is limited when learning a new task from scratch. In contrast, we only require the initial policy to sufficiently cover the state-action pairs often visited by an optimal policy. This is a much milder assumption and is reflected in our Experiment section, where a very weak initial LLM policy can still be useful. We are also focusing more on leveraging the LLM in-context decision-making capability to zero-shot boost the performance in RL tasks, which is critical in problems with high environment interaction cost or those with safety as a main concern.

**Theoretical analysis on LLMs' exploration in MDPs.** Recently, LLMs have shown very impressive capability Brown et al. (2020). Many works investigate how LLMs perform in in-context decision making compared to traditional methods, such as UCB, in MDP problems. For example, Arumugam & Griffiths (2025) introduces a more explicit method for exploration using Posterior Sampling. Chen et al. (2024) uses LLMs to construct multiple policies and combine with a model selection algorithm to solve Contextual Bandit. Lin et al. (2023) provides a theoretical framework to analyze supervised learning for in-context reinforcement learning. (Nie et al., 2024; Krishnamurthy et al., 2024) investigate how LLMs explore in the Bandit problem and show that the base LLM policies are non-trivial, but sub-optimal. This assessment aligns with our experiment results.

Many other works focus on **Embodied LLM and environment interactions** and **Using LLM to provide extra information for RL**, which we review in the extended related work Section A. Even with these successes, there are still many challenges in deploying LLM to solve sequential decision problems in practice, such as the lack of a guarantee of finding the optimal solution.

## 3 PRELIMINARIES

Consider a Markov Decision Process $M = (\mathcal{S}, \mathcal{A}, R, P, d_0)$, where $\mathcal{S}$ is the state space, $\mathcal{A}$ is the action space, the reward function is $R(s, a) \in \Delta([0, 1])$ and the transition dynamic $P(s, a) \in \Delta(\mathcal{S})$ at $(s, a)$, and $d_0(\mathcal{S}) \in \Delta(\mathcal{S})$ is the initial distribution. In this setting, the learner faces the MDP $M$ with $T$ episodes of horizon $H$. At each step $h$ of episode $t$, the learner chooses from its policy $\pi$ an action $a_h^t \sim \pi(s_h^t)$ and receives the reward from the reward function: $r_h^t = R(s_h^t, a_h^t)$, and transitions to the next state $s_{h+1}^t \sim P(s_h^t, a_h^t)$. The optimal policy $\pi^*$ is defined as a policy that has a maximum expected cumulative reward: $\pi^* = \arg\max_\pi \mathbb{E}\left[\sum_{t=1}^T \sum_{h=1}^H r_h^t \mid \pi\right]$.

We also have access to an initial policy $\pi_{\text{LLM}}$, with a limited query budget of $\tau$, the number of episodes we can query the LLM, that satisfies the coverage Assumption 1. Our goal is to maximize the cumulative reward by making use of $\pi_{\text{LLM}}$ to improve the sample efficiency.

**Assumption 1.** *We define the coverage upper bound to characterize the coverage property ($\downarrow$ is better):*

$$D_\pi = \sup_{h,s,a} \frac{d_h^\pi(s, a)}{\nu_h(s, a)},$$

*where $d_h^\pi(\cdot)$ is the state-action visitation distribution at step $h$ by following policy $\pi$ and $\nu_h(\cdot)$ is the state-action visitation distribution of the offline dataset. Then, following $\pi_{\text{LLM}}$ can produce trajectories with state-action pairs that sufficiently covers an optimal policy of the MDP.*

Assumption 1 states that an LLM can zero-shot suggest non-trivial base policies even when they are not optimal. We see an analogous phenomenon with the distribution shift problem in offline RL that results in a suboptimal policy in the online phase. Thus, we hypothesize that aggregating trajectories collected with LLM, which avoids trivial state-action data (such as unnecessary repetitions, visiting absorbing states, etc), and refining the learned policy later with online interaction, as the offline-to-online protocol, can be useful. This motivates our Assumption 1, which is known to allow offline-to-online RL to learn a high-performing policy (Song et al., 2022).

We provide empirical evidence on the CliffWalking and FrozenLake environments in Section 5.2, showing that the LLM collected data has much better state-action coverage, to justify Assumption 1.

## 4 THE LLM OFFLINE, RL ONLINE (LORO) ALGORITHM

Under Assumption 1, the policy $\pi_{\text{LLM}}$ collects high-quality data from the region that an optimal policy often visits. By only focusing on this and not exploring the low-quality data regions that are avoided by all optimal policies (*e.g.*,

---

**Algorithm 1** LLM Offline, RL Online (LORO)

---

1: **Input:** # of episodes $T$, # of LLM data collection episode $\tau$, episode length $H$, off-policy RL algorithm $\text{Alg}(\cdot)$
2: **Initialize:** LLM "offline" dataset: $\mathcal{D} = \{\emptyset\}$
3: **for** $t = 1, \cdots, \tau$ **do**                                                                                             ▷ LLM data collection
4:     **for** $h = 1, \cdots, H$ **do**
5:         Observe state $s_h^t$, take action $a_h^t \leftarrow \pi_{\text{LLM}}(s_h^t)$, and receive reward $r_h^t$
6:         Aggregate data $\mathcal{D} \leftarrow (s_h^t, a_h^t, r_h^t)$
7:     **end for**
8: **end for**
9: Pre-train the policy $\pi_{prev} \leftarrow \text{Alg}(\mathcal{D})$
10: **for** $t = \tau + 1, \cdots, T$ **do**                                                                                       ▷ Online learning
11:     **for** $h = 1, \cdots, H$ **do**
12:         Get policy $\pi_h^t \leftarrow \text{Alg}(\mathcal{D}, \pi_{prev})$                            ▷ Online updating the policy with the new data
13:         Observe state $s_h^t$, take action $a_h^t \leftarrow \pi_h^t(s_h^t)$, receive reward $r_h^t$
14:         Aggregate data $\mathcal{D} \leftarrow \mathcal{D} \cup \{(s_h^t, a_h^t, r_h^t)\}$
15:         Update $\pi_{prev} = \pi_h^t$
16:     **end for**
17: **end for**

---

hitting the wall, absorbing states, etc.), we can significantly improve the sample efficiency. In light of this, our LORO algorithm uses $\pi_{\text{LLM}}$ to collect a small amount of "offline" data to pretrain a policy $\pi$ with it, and then use a classical online learning algorithm to fine-tune $\pi$ to be optimal with a much smaller number of observations.

We present the details of LORO in Figure 2 and Algorithm 1. Initially, we use the LLM policy $\pi_{\text{LLM}}$ to collect data for the first $\tau$ episodes (line 3 - 8). Then, we pre-train a policy using an off-the-shelf off-policy RL algorithm on the data collected by LLM (line 9). Finally, we online fine-tune the pre-trained policy (line 10 - 17).

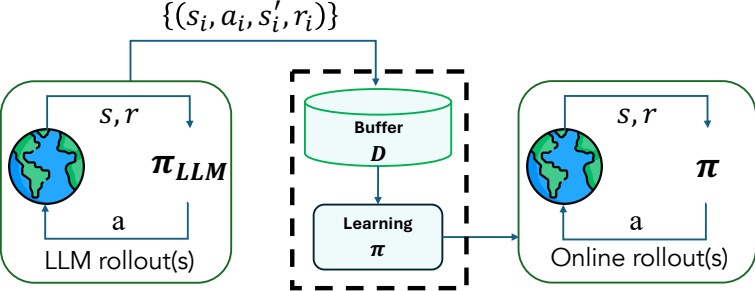

Figure 2: The LLM Offline, RL Online (LORO) algorithm. Image inspired by Levine et al. (2020).

## 5 EXPERIMENTS

We empirically evaluate our algorithm on a host of RL environments: Cart Pole, Pendulum, Frozen Lake, Cliff Walking, Represented Pong, and Mountain Car. We defer the environments' descriptions and RL implementation details to Appendix B, and the LLM setup to Appendix G.

Here, we compare our algorithm with the following baselines:

- **Online RL**: an off-the-shelf RL method that collects the data and refines its policy in an online manner. In particular, we use Double-DQN van Hasselt et al. (2015) for discrete action environments and SAC Haarnoja et al. (2019) for continuous action environments.

- **LLMs as Policies (Qwen-7B-Instruct, Qwen-32B-Instruct)**: the base policies from the 7B and 32B of the Qwen 2.5 series with Instruction tuning Yang et al. (2024). For each episode $t$ and step $h$, the LLM has access to the environment and observation descriptions $s_h^t$, and the action $a_h^t$ is taken using Chain-of-Thought Wei et al. (2022). The LLM setup details are in Appendix G. The prompt setup and examples are in Appendix H.

Note that we only show the average episode reward collected in the first $\tau$ episodes, $r_{\text{avg}} = \frac{1}{H\tau} \sum_{t=1}^{\tau} \sum_{h=1}^{H} r_h^t$, in the figures below.

- **Random**: a policy $\pi_{\text{random}}$ that take action $a_h^t$ uniformly at random. Similarly, we only show the average episode reward collected in the first $\tau$ episodes.

In the experiments below, we choose $\tau = 10$ and the number of pre-training steps is 1000. The task length $T$ is 150 for CartPole, FrozenLake, 200 for CliffWalking, Pendulum, RepresentedPong, and 300 for MountainCar. LORO is trained using the data collected by Qwen-7B. To thoroughly evaluate the effectiveness of LORO and understand the underlying factors influencing its performance, we organize our experiments into several parts: we begin by demonstrating the main results (§5.1) across diverse RL environments, verify Assumption 1(§5.2), then conduct a series of ablation studies to assess the impact of pre-training (§5.3), the quality of LLM-collected data (§5.4), and the LLM's reasoning capabilities (§5.6). We also verify LORO's agnosticity to base RL learner choice in §5.5. In addition, we explore the effects of LLM model size, and the number of pre-training steps, and varying the amount of LLM data $\tau$, with results provided in Appendix E.2, E.3, and E.4. The wall-clock time for running the experiments is provided in Appendix F.

## 5.1 SAMPLE EFFICIENCY

The main results of our algorithm are shown in Figure 3. In all learning curves, the first $\tau = 10$ episodes in LORO show the average episode reward using the pure LLM-based policies. Afterwards, LORO significantly outperforms the LLM-based policies and the Online RL baselines in four environments. Notice that the base LLM policies are often not optimal, but they can still generate high-quality trajectories to improve the sample efficiency of LORO, which justifies Assumption 1.

We also highlight that the drop in LORO's performance at episode 10 in the RepresentedPong environment is due to an improper choice of $\tau$ (the number of episodes running with an LLM policy). Similarly, at episode 10, we can see a step in LORO's performance in the other environments. Ideally, we want to keep using LLM until the RL policy achieves at least a similar performance to the LLM policy.

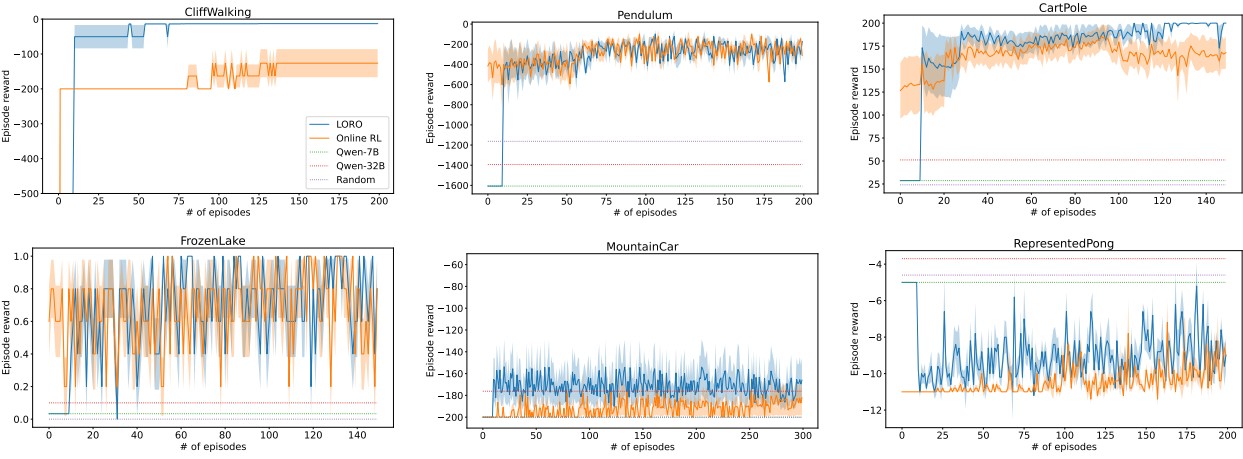

Figure 3: Our algorithm, LORO, outperforms the LLM policies (Qwen 7B, Qwen 32B) and the Online RL baselines in four environments and competitive in the other two. LORO learns the optimal policy in the first four environments, while the Online RL only does so for three. Even when not converged to the optimal solution, LORO outperforms other baselines in the last two more difficult environments. LLM's base policies can perform very well in the RepresentedPong experiment, maybe because of the extra information that we provide for this setting, described in Appendix B. All results are shown with one standard error over five random seeds. In the CliffWalking experiment, some baselines are not shown in the figure since their episode rewards are too small (-509 for Qwen-7B and -7729 for Qwen-32B). Similarly, Qwen-7b and random overlap at -200 on the MountainCar experiment.

| Task | LORO | Online RL | Qwen-7B | Qwen-32B | Random |
|------|------|-----------|---------|----------|--------|
| CliffWalking | **1** | 2 | 3 | 5 | 4 |
| Pendulum | 2 | **1** | 5 | 4 | 3 |
| CartPole | **1** | 2 | 4 | 3 | 5 |
| FrozenLake | 2 | **1** | 4 | 3 | 5 |
| MountainCar | **1** | 3 | 4 | 2 | 4 |
| RepresentedPong | 4 | 5 | 3 | **1** | 2 |
| **Avg. Rank** | **1.83** | 2.33 | 3.83 | 3.00 | 3.83 |

Table 1: Final cumulative rewards ranking ($\downarrow$ is better). Best per task in **bold**. The full results is in Table 4.

| "Offline" dataset (30 episodes) | # missing good state-action $\downarrow$ | Surrogate transfer coef upperbound $\tilde{C}_{\pi'}$ (Eq. (1)) $\downarrow$ |
|------|------|------|
| Qwen 7B | $0.00 \pm 0.00$ | $69.10 \pm 0.00$ |
| Qwen 32B | $0.00 \pm 0.00$ | $1279.17 \pm 2524.68$ |
| Online collected | $0.00 \pm 0.00$ | $25.65 \pm 34.73$ |
| Random collected | $0.80 \pm 0.45$ | $\infty$ |

Table 2: The Surrogate Transfer Coef upperbound, in Eq. (1), approximates the Transfer Coefficient upperbound from Song et al. (2022), which measures the coverage between the CliffWalking's offline dataset and an optimal policy ($\downarrow$ is better). Data collected by LLM has a very low missing state-action. Even though the Online and Random collected data can seemingly have good coverage sometimes, as demonstrated in Figure 1 and 4, this was due to the fact that the final policy is not optimal. The full table for both CliffWalking and FrozenLake environments is shown in Table 3.

## 5.2 Justifying the Coverage Assumption 1

In this section, we verify Assumption 1 on whether LLM's collected data covers more state-action spans by an optimal policy compared to data collected by a vanilla Online RL or Uniform Random policy. Song et al. (2022) provides an upper bound of the Transfer Coefficient of a policy: $C_\pi \leq D_\pi$. Here, since there can be more than one optimal policy, we assume one can be found after the online learning process and evaluate $\pi$, the policy after online training, as an approximation of the optimal policy. Because the offline dataset is too small in our setting, replacing the terms in the density ratio with sample-based estimates of $d_h^\pi(s, a)$ and $\nu_h(s, a)$ would make the ratio infinite. Hence, we define $d^\pi(s, a)$ as the state-action visitation distribution by following policy $\pi$ (not limited to a specific step $h$ as the original formulation). Similarly, $\nu(\cdot)$ is the state-action visitation distribution of the offline dataset. Thus, we define the surrogate transfer coefficient upper bound:

$$\tilde{C}_\pi := \sup_{s,a} \frac{\tilde{d}^\pi(s, a)}{\tilde{\nu}(s, a)}, \tag{1}$$

where $d_h^\pi(\cdot)$ and $\nu_h(\cdot)$ are the state-action visitation distributions of policy $\pi$ and of the offline dataset. We estimate $\tilde{C}_\pi$ using data; specifically, we replace $d^{\pi'}(s, a)$ and $\nu(s, a)$ with their sample-based counterparts.

In Table 2, "# missing state-action" is the number of state-action pairs that appear in the final policy, which approximate an optimal policy, but do not appear in the "offline" dataset. It is a good indicator of the finiteness of the surrogate transfer coefficient upper bound $\tilde{C}_{\pi'}$ in Eq. (1). Since the data collected by LLM have very small values, this justifies our assumption. We further demonstrate the policy's state-action visitation traces in Figures 1 and 4, where the data collected by an LLM overlaps with the optimal policy much more than the other baselines, to show that the transfer coefficient maps correctly to the optimal policy coverage, unlike Online and Random data collection.

In Figure 1 and 4, the "offline" dataset (red) is collected from the first ten episodes. Similarly, the final policy's trajectories, after finishing learning, are collected for ten episodes and shown in the figures. The details of this section, including the results for both the FrozenLake and CliffWalking environments, are provided in Section C.

When Assumption 1 is violated, our experiments show that warm-starting with LLM-collected data remains robust, typically outperforming alternative baselines across diverse environments. By contrast, even when the assumption fails for the Online RL and Random data regimes, sufficient exploration during the subsequent online phase can sometimes compensate, yielding competitive performance in some settings—albeit usually with higher sample complexity.

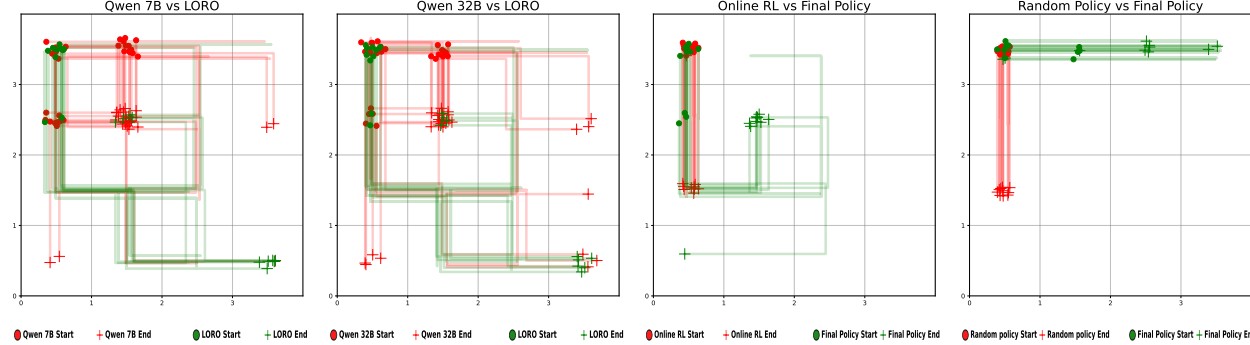

Figure 4: Traces in the FrozenLake environment. Similar to the CliffWalking environment in Figure 8, the LLM collected data covers much better the state spaces compared to Online and Random collected data.

## 5.3 EFFECT OF PRE-TRAINING

In this section, we aim to verify the importance of pre-training (Algorithm 1 line 9) is for LORO's performance through an ablation study, since Song et al. (2022)'s algorithm only mixes the offline and online data without pre-training to avoid being too conservative toward the pre-training data, which may hurt the performance.

In Figure 5, we show that mixing offline and online data alone (which is equivalent to Song et al. (2022)) is insufficient. Our conjecture is that the LORO's pretraining step trains a good policy using only the high-quality data without the data from regions less visited by the optimal policy. As shown in Figure 3, pre-training significantly boosts the performance of LORO compared to just mixing the LLM's collected data with the online RL collected data after $\tau$ episodes.

Even though pre-training can be useful initially, to behave optimally, the agent still needs to explore other state-action pairs in case the initial data comes from a sub-optimal policy, as shown in the CartPole environment in Figure 3.

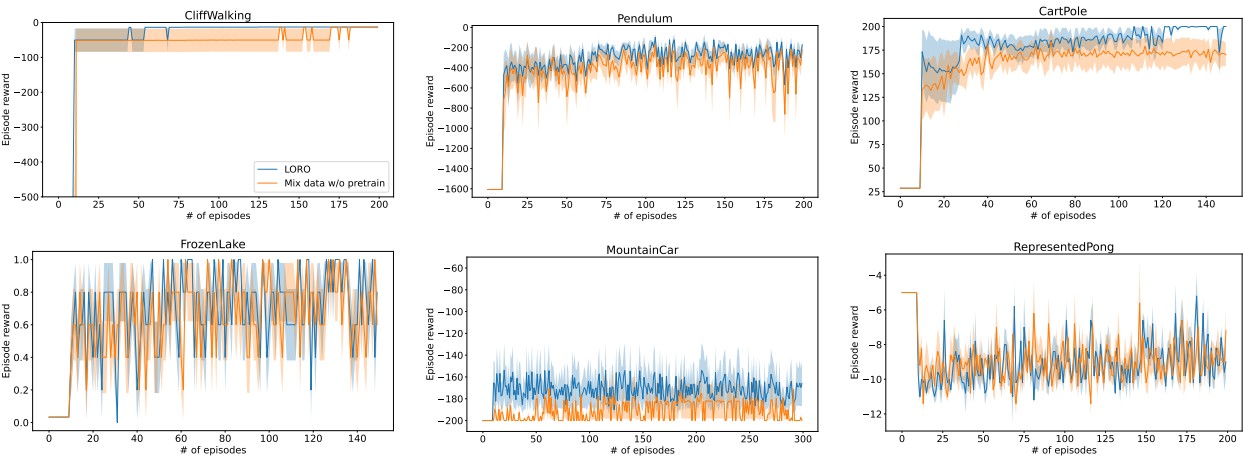

Figure 5: Comparing pre-training (then removing the collected data) versus mixing the LLM's collected data with online RL data without pre-training. It's clear that pre-training is necessary for LORO to achieve superior performance compared to naively mixing the data.

At first sight, there seems to be a contradiction between our findings and Song et al. (2022). They assume access to a large offline dataset and, along with Nair et al. (2020), want to keep the policy less conservative toward the offline data by treating the online versus offline data equally. In contrast, we don't have access to offline data. We instead use LLM to collect a small number of high-quality data, thus, unlike Song et al. (2022), LORO has a higher updates-to-data ratio for these observations. Our experiment shows that being conservative by "overfitting" to the LLM dataset can help learning more efficiently.

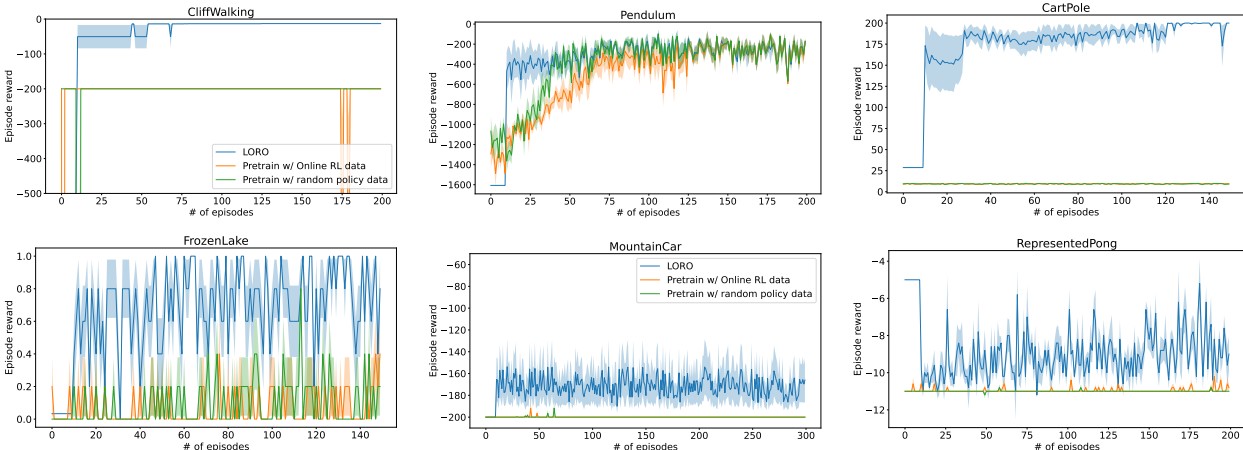

Figure 6: Comparing pre-training with LLM's data versus random and online RL data. The main finding here is that pre-training is only useful with LLMs' data.

In addition, we may question whether pretraining accumulates primacy bias Nikishin et al. (2022) since we are "overfitting" in earlier observations and hurt the performance. Since pretraining is helpful in our experiment in Figure 5, even in difficult environments such as MountainCar and RepresentedPong, the effect of primacy bias is insignificant in our experiments. Furthermore, when applying the same pretraining steps for different collected datasets, possibly exposing them all to primacy bias, LORO still outperforms all other baselines as shown in Figure 6. This suggests that the quality of the collected dataset is the primary reason for LORO's performance.

Combining LORO with the resetting trick in Nikishin et al. (2022) to address the primacy bias is a simple task. Similarly, we can easily combine the high-quality data of LORO to many algorithms to make good use of the LLM-collected offline dataset, as shown in the Related Work section. Since this is out of scope for this paper, we evaluate the simplest approach of directly pretraining with the pseudo-offline dataset to highlight the effectiveness of the LLM policy for warm-start learning. Further evaluating the usefulness of LLM-collected data in different offline-to-online algorithms is an interesting open question that we left for future work.

## 5.4 EFFECT OF LLM'S DATA

In the previous section, Figure 5 shows the importance of pre-training using data collected by LLM. In this section, we perform an ablation study that demonstrates that the quality of such data is crucial. In Figure 6, we show that using LLM's collected data is significantly better than using data collected with an Online RL algorithm from scratch or a policy that takes actions uniformly at random. Thus, we conclude that pre-training is only beneficial when coupled with high-quality data, which supports our conjecture above.

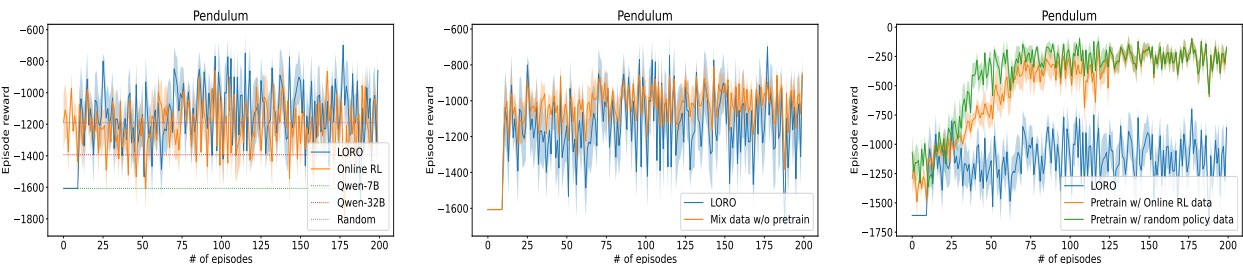

Figure 7: Comparing the effect of LLM collected data versus Online or Random collected data using AWAC Nair et al. (2020) as the base algorithm. The first figure shows the general sample efficiency comparison, the second compares with the mix data baseline, and the third compares with different pre-training datasets. LORO uses Qwen-32B for the offline phase in this experiment.

## 5.5 Verifying LORO's agnosticity to base RL learner choices

In this section, we aim to verify whether the performance boost from using LLM is truly algorithm-agnostic. Here, we show the results from using AWAC Nair et al. (2020) as the main algorithm and comparing its performance when using LLM collected data versus Online or Random collected data. The results are shown in Figure 7. We can see that LORO still outperforms most baselines, with the exception of the Pretrain with Online or Random baselines. Further experiment results using AWAC are shown in Section E.1. Understanding why pretraining is useful when combined with a small amount of high-quality data collected by an LLM is an interesting question left for future research.

## 5.6 Effects of the LLM's capability

Given that the performance of many reasoning tasks increases with the improvement of the LLM's capability through: increasing the model's size, using Supervised Fine-Tuning (SFT) Ouyang et al. (2022), Long Chain-of-Thought (CoT) Chen et al. (2025), or some Test-time-scaling methods such as Majority Voting Wang et al. (2022), and Best-of-N Cobbe et al. (2021). In this section, we want to investigate whether this increase in LLM's reasoning capability also translates to decision making in MDP problems.

From our experiments in Appendix E.2, we see no clear link between an LLM's model size and its performance. On the other hand, we notice that the LLM's base policies are only useful using CoT instead of just asking the LLM to make decisions. We also observe that the 0.5B model is not useful, as well as using Majority Voting or Best-of-N without CoT. Hence, in Appendix E.5, we investigate if increasing the LLM's capability using SFT or using an LLM with Long CoT can help. We show that there is no significant difference in using standard CoT compared to using SFT or Long CoT. Understandably, SFT wouldn't be useful, or may even be counter-productive, since the amount of data collected for fine-tuning is too small to make a difference (around 500-3000 prompt-response pairs in our experiments). Hence, we conclude that improvements over LLM's capability do not directly translate to improvement in warm-starting RL tasks.

## 5.7 Other findings

Interestingly, we find that a small model size (7B) is more sensitive in a few environments , such as MountainCar (with 3000 pre-training steps in Figure 15) and FrozenLake (with bad history summarization in Figure 32). We also find that the amount of pre-training data in general does not affect the learned policy's cumulative return, which is shown in Appendix E.4.

Besides what we reported above, we find no clear relationship between the task's performance and the number of pre-training steps or the model size. These are shown in Appendix E.3, E.2. We also found no clear difference between environments with Discrete Action versus Continuous Action (e.g., Pendulum), despite the intuition that the Discrete Action environments should be easier for the LLM Singh et al. (2025).

## 6 Conclusion and Future work

In this paper, we investigate how to leverage an LLM to warm-start traditional RL methods. Empirically, we have shown that the high-quality data collected by the LLM can significantly increase the sample efficiency of online RL. Our definition of data quality follows from previous work Song et al. (2022). Our algorithm further utilizes pretraining to take full advantage of LLM-collected data.

Our work provides a framework for significantly reducing the sample complexity in RL problems. This is especially important for practical applications where the data collection cost or safety is a major concern. A limitation of our work is that Assumption 1 may not hold for some RL tasks, but we believe that the increasing capability of LLM would increase the range of problems where Assumption 1 is applicable. In the future, we would like to extend this work to more sophisticated RL problems, with a large State and Action space. We would also like to investigate how to scale the sample efficiency with the LLM's capability and how to choose the LLM query budget $\tau$ in a principled manner.

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

# A   Extended related work

**Offline-to-online RL:** Many more sophisticated approaches have been studied empirically. However, none of these works studied the utility of LLMs in warm-starting online RL. Ball et al. (2023), in particular, proposes that distribution shift exacerbate the problem of bootstrap error propagation in off-policy methods, especially with a function approximation like a Neural Network. Hence, they suggest multiple practical tricks such as: balance sampling the offline and online data, using LayerNorm Ba et al. (2016) and Clipped Double Q-Learning Fujimoto et al. (2018) to reduce instability while avoid excess conservatism.

**Embodied LLM and environment interactions:** Recently, LLM has showed very impressive capability Brown et al. (2020), including understanding about physics Patel & Pavlick (2022) Liu et al. (2024), color Abdou et al. (2021), and affordances between bodies and object Ahn et al. (2022). This implicit knowledge could be the reason why LLM can be used to directly manipulate robots Ahn et al. (2022), Huang et al. (2022), Liang et al. (2023). However, Carta et al. (2023) claims that LLMs lack grounding due to 1) the training objective of next word prediction not aligned with other goals, and 2) no interactions with the environment.

Many works seemingly agree with Carta et al. (2023) and incorporate environment interactions, thus showing significant improvement. A popular approach is letting the LLM interact directly with the environment and collect the feedback for the subsequent prompt Carta et al. (2023), Yao et al. (2022), Zhou et al. (2023), Luketina et al. (2019). Another direction is a two level system, where the LLM take high level, abstract actions (such as creating sub-goals Bhat et al. (2024) Dalal et al. (2024) or choosing the skills to use Liang et al. (2023), Ahn et al. (2022)), and the low level classical system implementing the LLM's "plan" in practice. A related work from Hao et al. (2024) uses LLM to extract and formulate the problem's objectives, constraints, and may include sub-goals creation, for the low-level optimization solver.

**Using LLM to provide extra information for RL:** Carta et al. (2023) and Tan et al. (2024) use LLM directly to generate the policy and fine-tune it with RL (using Policy Gradient with PPO or an Actor-Critic framework). Lee et al. (2023) and Lin et al. (2023) propose pretraining an LLM with an offline dataset and show that it can both explore online and act conservatively offline. Unlike them, instead of an end-to-end approach that mixes the RL objective (of maximizing the cumulative reward) with the LLM objective (for next token prediction), we have a separate, smaller RL learner trained exclusively on the classical RL objective that enjoys the typical asymptotic optimality Since one of our motivations is computational efficiency, hence, training a large neural network that requires a lot of data would defeat the point of using LLM to help reduce the sample complexity. Yan et al. (2024) uses LLM to provide the action prior, then train a policy to do posterior sampling using it. Zheng et al. (2022) pre-trains a transformer-based neural network on the offline dataset and develops a way to efficiently fine-tune it with online interaction. This differs from our proposal since we don't have an offline dataset, but the data collected by the LLM's policy can be regarded as a small offline dataset. Furthermore, our work is algorithm agnostic. We propose that the data collected by LLM is of high-quality and can later be used to train a policy using other algorithms. Another closely related work is Du et al. (2023), where the LLM guides the algorithm's exploration by generating (sub) goals and rewards the RL algorithm when achieving these goals. Similarly, Bhambri et al. (2024) uses heuristics from LLM to combine with RL in the Potential Based Reward Shaping framework. While these works and ours leverage LLM to reduce unnecessary exploration for RL, they focus more on sub-goal generation and providing intrinsic reward in sparse feedback problems, while we are focusing on dense reward settings where RL online interactions can refine the warm-started but sub-optimal policy given by the LLM. Finally, Choi et al. (2022) and Kant et al. (2022) use LLM to provide a prior for the policy to help the learner explore more efficiently, which is similar to our motivation on a high level.

**Other ways LLM can help solving MDPs:** Besides low-level control and high-level planning, Jeong et al. (2024) also investigates how LLM can help robot intelligence systems by reward design (to combine with RL) Ma et al. (2023), Xie et al. (2023), and scene understanding Huang et al. (2023), Hong et al. (2023). Even with these successes, there are still many challenges in deploying LLM to solve sequential decision problems in practice, such as the lack of a guarantee of finding the optimal solution.

# B   The environments and implementation details

## B.1   The environments

We empirically verify our algorithm on some classic RL environments:

- **Cart Pole**: The agent aims to balance a pole on top of a cart by moving left and right. It observes the Cart Position, Cart Velocity, Pole Angle, and Pole Angular Velocity. The reward is one for every step taken before the episode ends, either by having the pole fall over, moving the cart to the edge of the display, or reaching the maximum episode length.

- **Pendulum**: The agent aims to swing up an inverted pendulum by applying torque on its free end. It observes the $(x, y)$ location of the pendulum's free end and its angular velocity. From the location, we calculate the pendulum's angle and the rotating direction to help the LLM, but do not use them in the online phase. The reward is calculated based on the pendulum angle, where the upright location has the highest reward. The episode ends when it reaches the maximum episode length. Note that the action set here is continuous, which can be more challenging for the LLM's policy.

- **Frozen Lake**: The agent aims to move from the top-left to the bottom-right location in a four-by-four grid world. The agent can move up, down, left, and right. It only observes its own location. The reward is zero everywhere except at the goal, where the reward is one. The episode ends either when the agent moves to one of the four "holes" in the grid, reaches the goal, or reaches maximum episode length. We further implement an external environment history to store the rewards received at each visited location, which is necessary for the LLM to solve this task. The environment's history is not used in the online learning phase.

- **Cliff Walking**: The agent aims to move from the bottom-left to the bottom-right location in a four-by-twelve grid world. The agent can move up, down, left, and right. It only observes its own location. The reward is negative one everywhere except negative one hundred at the cliff locations on the bottom of the grid. The episode ends either when the agent reaches the goal or reaches maximum episode length. We also use the environment history for this environment.

- **Represented Pong**: This is the Atari game Pong, but instead of the traditional image observation, we use Anand et al. (2019) to extract the game state information from the RAM state. The agent then observes its own coordination, the ball's, the opponent's, and the score. We also calculate the ball velocity and add it to the observation, since it seems necessary to ensure Markov's property (able to take optimal action with only the current state information). The agent controls the right paddle up and down and competes against the left paddle controlled by the computer by trying to deflect the ball away from your goal and into the opponent's goal. The agent receives a point whenever it scores a goal and loses when the opponent does. The game ends when a player's score reaches twenty-one or the agent reaches the maximum episode length.

- **Mountain Car**: The agent's goal is to move from the bottom of a sinusoidal valley to the top of the right hill as quickly as possible. The agent can strategically accelerate left or right. It only observes its location and velocity. The reward is negative one everywhere except the goal. The episode ends either when the agent reaches the goal or reaches maximum episode length.

## B.2 IMPLEMENTATION DETAILS

We build our code from Pandey (2024), which provides a framework for LLM interacting with OpenAI's gym games with a built-in text description wrapper to turn RL games into something LLM can play. The game descriptions, which are listed in Appendix H, are heavily referenced from K (2024). The RL training process is using d3rlpy Seno & Imai (2022), with the default hyperparameter choice, with batch-size 256, buffer size 100,000, $\varepsilon : 0.1$, $\gamma : 0.99$, target update interval 1,000, and learning rate $5e - 5$. We use DDQN van Hasselt et al. (2015) for all tasks with Discrete Action and SAC Haarnoja et al. (2019) for Continuous Action. The LLM was run on two H100 GPUs.

For the LORO algorithm, we collected data to pre-train a policy and then only used online data in the online learning process. Even though we recommend following the Algorithm 1 and aggregate both the pretrain and online collected data, in our experiment, we remove the data collected in the "offline" phase after pretraining to have a clear ablation study on whether the improvement of LORO is coming from the mixing the offline versus online or does pretraining is also important, as shown in Figure 5 and Figure 6.

## C JUSTIFYING THE COVERAGE ASSUMPTION 1

Below, we show the full Surrogate Transfer Coefficient Table in Table 3. We also show the full traces of different algorithms and baselines in the CliffWalking environment in Figure 8.

| Environment | "Offline" dataset (30 episodes) | # missing good state-action ↓ | Surrogate Transfer Coef upperbound $\tilde{C}_{\pi'}$ (Eq. (1)) ↓ |
|---|---|---|---|
| CliffWalking | Qwen 7B | $0.00 \pm 0.00$ | $69.10 \pm 0.00$ |
| | Qwen 32B | $0.00 \pm 0.00$ | $1279.17 \pm 2524.68$ |
| | Online collected | $0.00 \pm 0.00$ | $25.65 \pm 34.73$ |
| | Random collected | $0.80 \pm 0.45$ | $\infty$ |
| FrozenLake | Qwen 7B | $7.60 \pm 0.55$ | $\infty$ |
| | Qwen 32B | $8.60 \pm 1.14$ | $\infty$ |
| | Online collected | $6.20 \pm 5.89$ | $\infty$ |
| | Random collected | $10.80 \pm 8.47$ | $\infty$ |

Table 3: The Surrogate Transfer Coef upperbound, in Eq. (1), approximates the Transfer Coefficient upperbound from Song et al. (2022), which measure the coverage between the offline dataset and an optimal policy (lower is better). "# missing state-action" is the number of state-action appears in the final policy, which approximate an optimal policy, but not appears in the "offline" dataset. Data collected by LLM has very low missing state-action. Even though the Online and Random collected data can seemingly have good coverage sometime, as demonstrated in Figure 8 and 4, this was due to the final policy is not optimal.

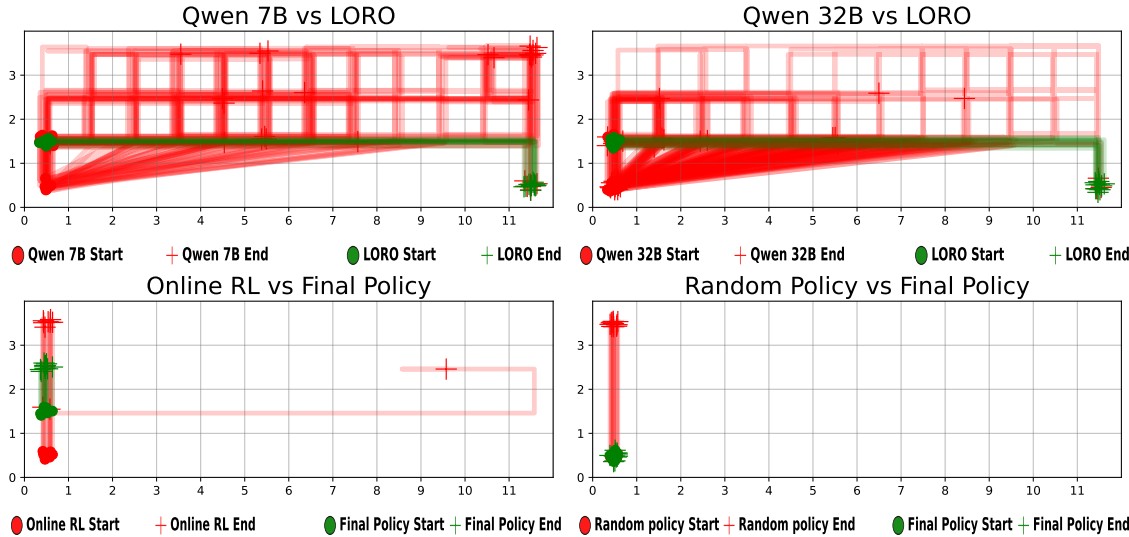

Figure 8: Traces in the CliffWalking environment. The red traces are trajectories collected in the first 10 episodes. The green traces, represent the traces of the optimal policy, collected by the evaluating the final policy after fine-tuning for 190 episodes. It's clear that the "offline" dataset (red) collected from a LLM covers the optimal policy much more than the ones collected from a vanilla Online RL or Uniformly Random policy.

| Task | LORO | Online RL | Qwen-7B | Qwen-32B | Random | Mix | Pretrain | |
|---|---|---|---|---|---|---|---|---|
| | | | | | | | Online RL | Random |
| CliffWalking | **-9213** | -37151 | -101933 | -1545806 | -404919 | -17449 | -49504 | -55840 |
| Pendulum | -71703 | **-58707** | -321416 | -278678 | -239960 | -91592 | -100974 | -82163 |
| CartPole | 26035 | 24640 | 4299 | 7675 | 3104 | **26599** | 1396 | 1398 |
| FrozenLake | 99 | **100** | 4 | 14 | 0 | 93 | 10 | 14 |
| MountainCar | **-51611** | -57616 | -60000 | -52859 | -60000 | -57238 | -59988 | -59984 |
| RepresentedPong | -1784 | -2088 | -1000 | **-740** | -920 | -1772 | -2191 | -2199 |
| **Avg. Rank** | **1.7** | **1.7** | 5.8 | 4.3 | 6.3 | 2.7 | 6.3 | 5.7 |

Table 4: Final cumulative rewards (↑ is better). Best per task in **bold**.

# D  CUMULATIVE REWARDS TABLE

In this section, we show the cumulative rewards of all algorithms and baselines in the main paper in Table 4.

# E  ABLATION STUDY

## E.1  ALGORITHM AGNOSTIC VERIFICATION

In this section, we show further effects of model size and the amount of the offline data to the final performance of LORO using AWAC Nair et al. (2020) in Figure 9 and Figure 10. Compared to the SAC performance in the Experiment section, AWAC gives worse performance overall and more sensitive to the LLM model size, since the Qwen-7B data performs much worse than Qwen-32B. Still, LORO using AWAC still outperforms the baselines.

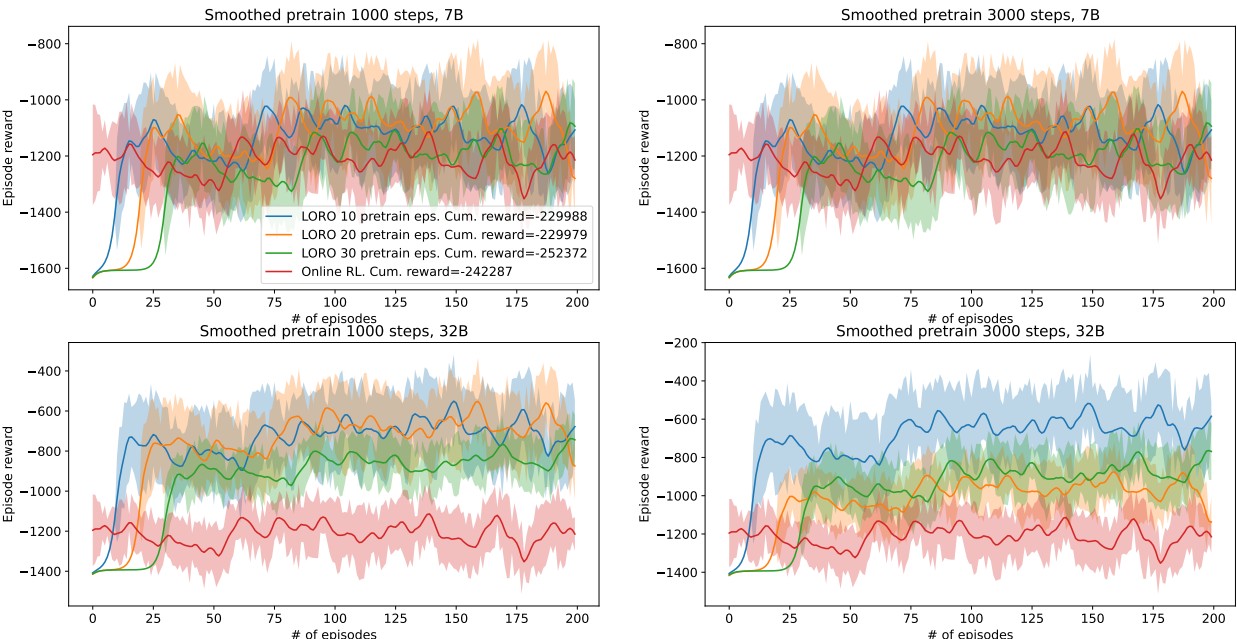

Figure 9: Comparing the effect of different LLMs' model sizes for the Pendulum environment with LORO using AWAC Nair et al. (2020).

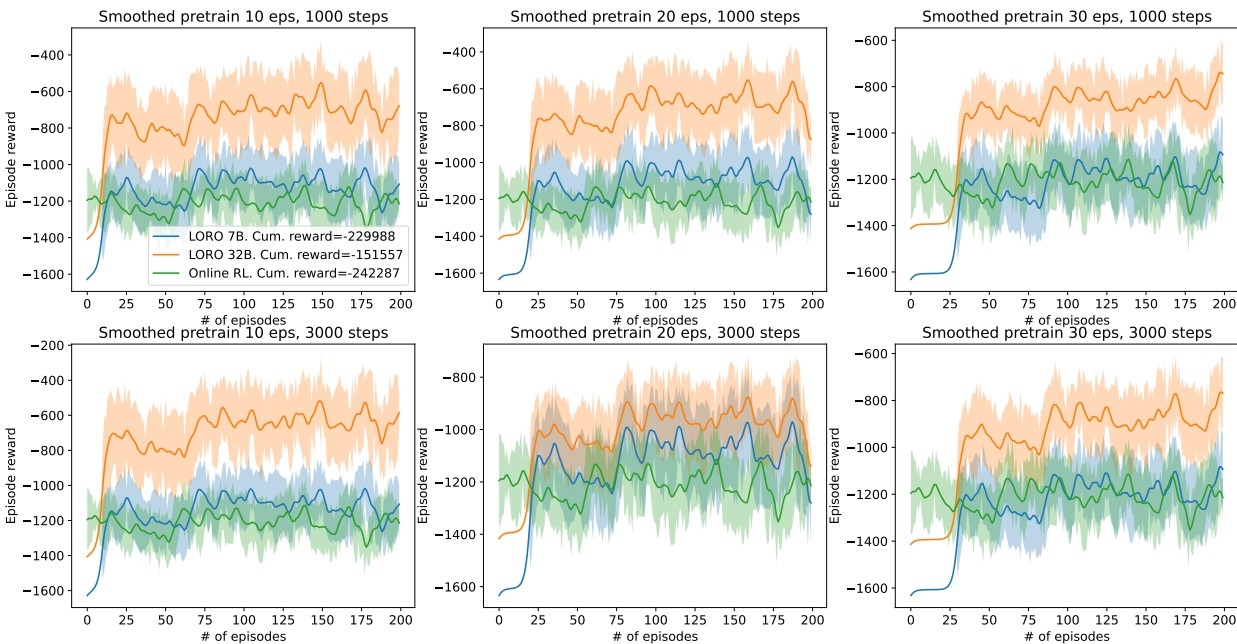

Figure 10: Comparing the effect of different LLMs' model sizes for the Pendulum environment with LORO using AWAC Nair et al. (2020).

## E.2    EFFECTS OF LLM'S MODEL SIZE

In this section, we evaluate the effect of the LLM's model size on the cumulative reward of the policy. We evaluate this with different pre-training data and pre-training steps on six OpenAI Gym environments and show the result in Figure 11, 12, 13, 14, 15, 16. Overall, we observe no clear advantage of using a larger model to improve the decision-making quality of the LORO policy.

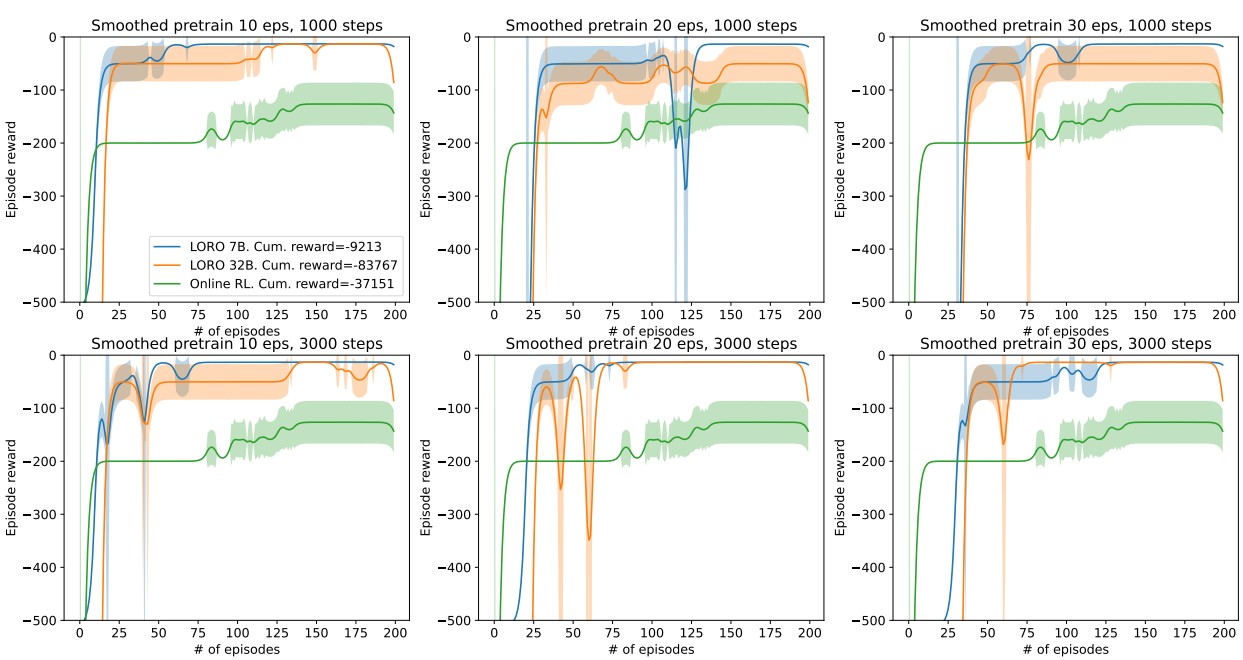

Figure 11: Comparing the effect of different LLMs' model sizes for the CliffWalking environment.

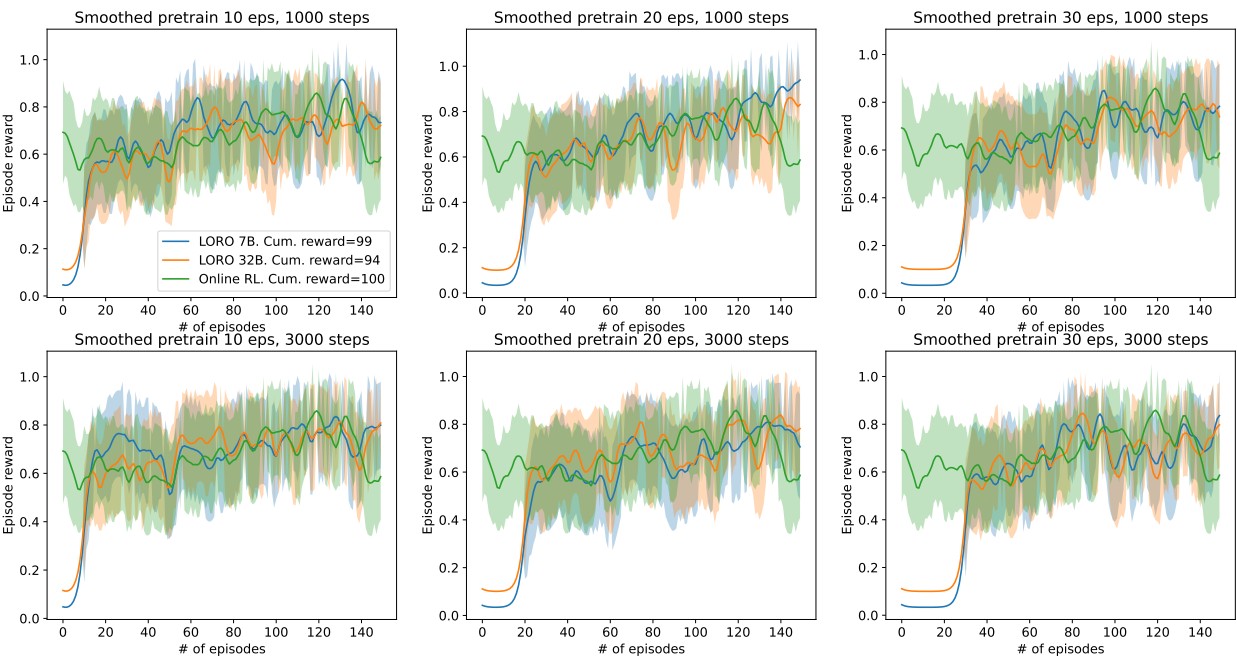

Figure 12: Comparing the effect of different LLMs' model sizes for the FrozenLake environment.

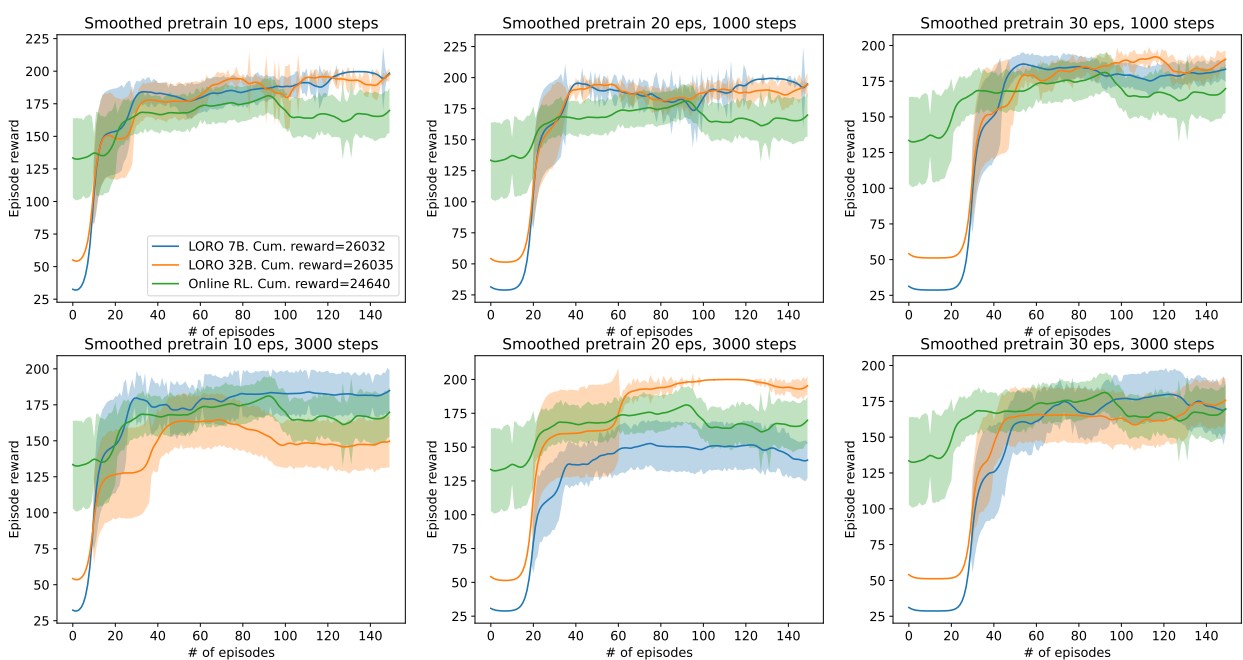

Figure 13: Comparing the effect of different LLMs' model sizes for the CartPole environment.

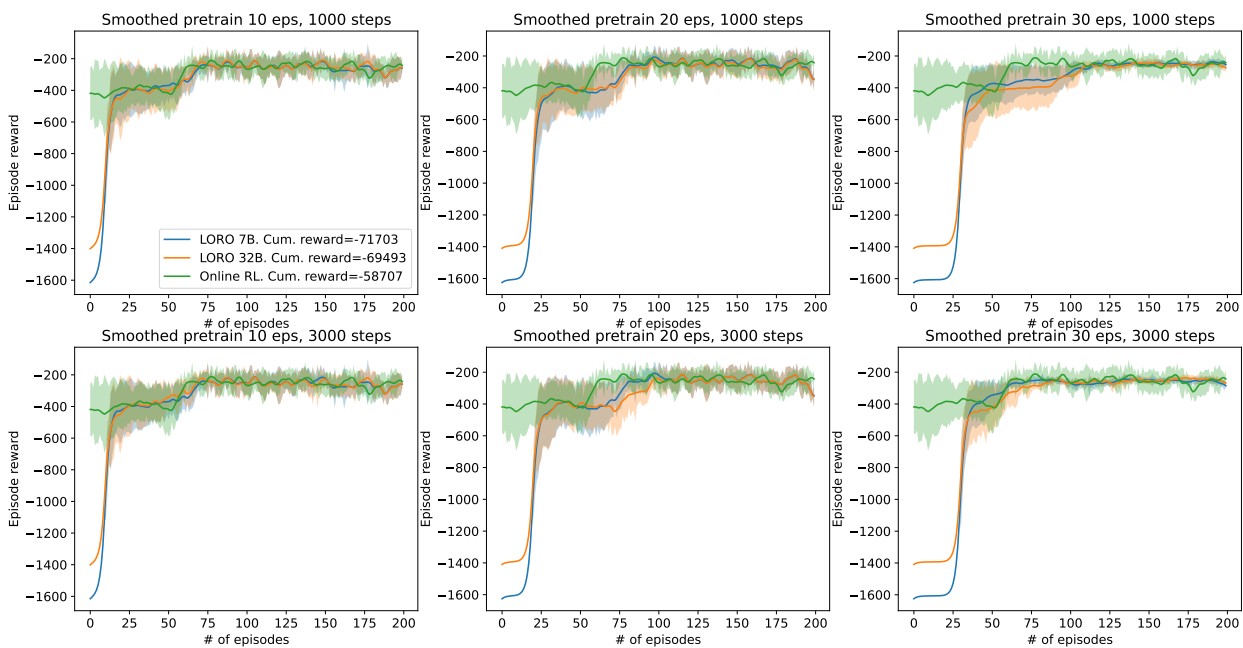

Figure 14: Comparing the effect of different LLMs' model sizes for the Pendulum environment.

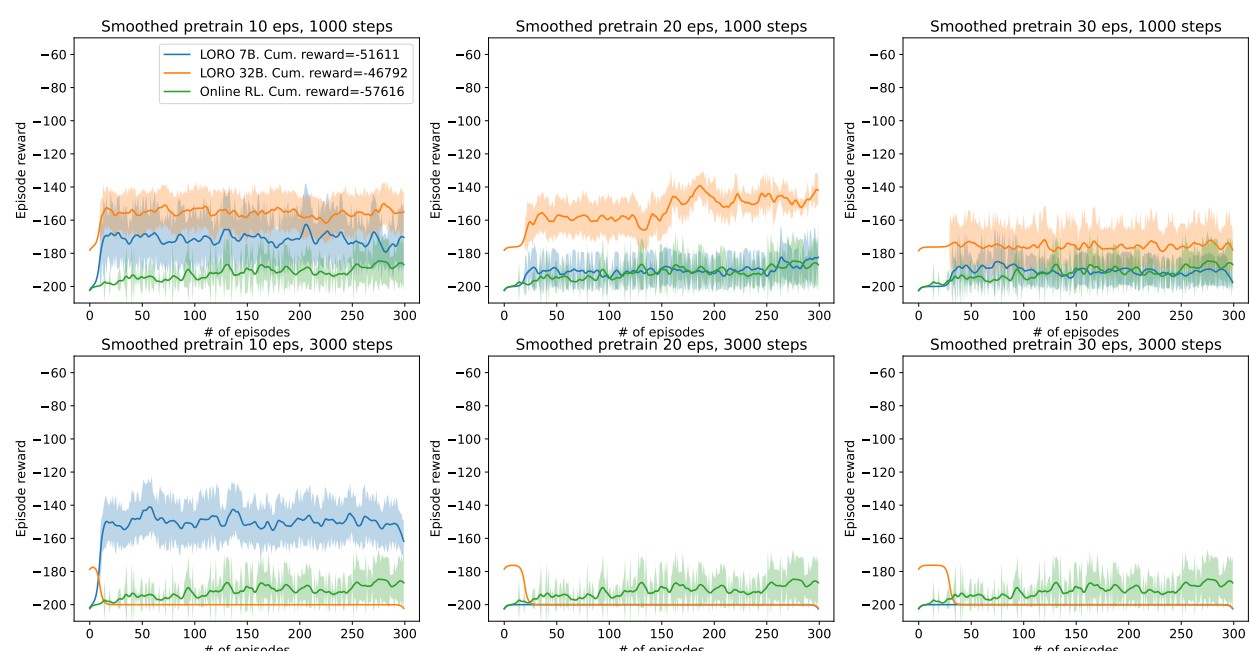

Figure 15: Comparing the effect of different LLMs' model sizes for the MountainCar environment.

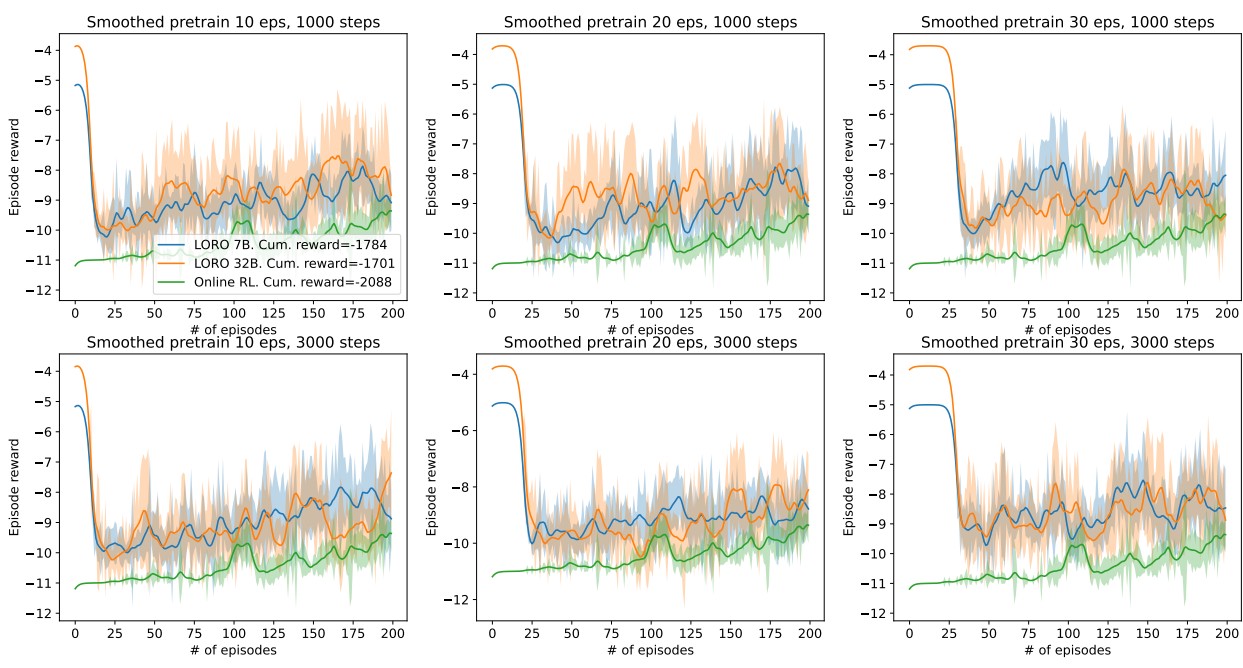

Figure 16: Comparing the effect of different LLMs' model sizes for the Pong environment.

### E.3 Effects of the number of pre-training steps

In this section, we evaluate the effect of the number of pre-training steps on the cumulative reward of the policy. We evaluate this with different model sizes and pre-training data on six OpenAI Gym environments and show the result in Figure 17, 18, 19, 20, 21, 22. Overall, we observe no clear advantage of using a higher or lower number of pre-training steps to improve the decision-making quality of the LORO policy.

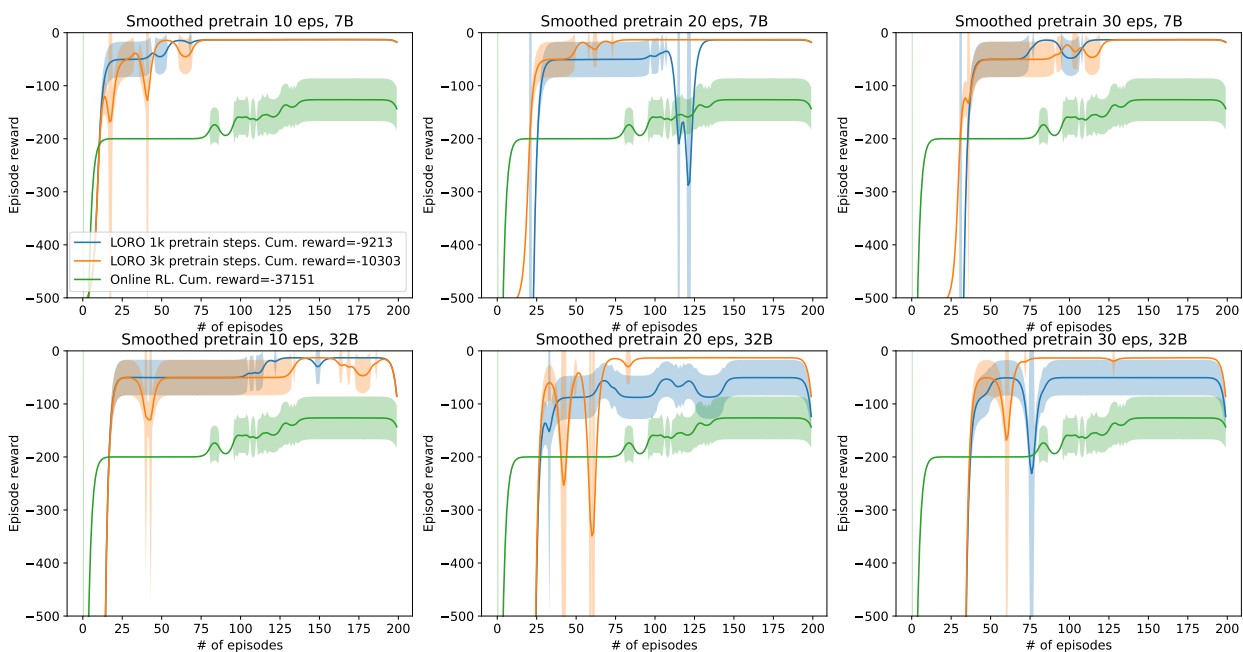

Figure 17: Comparing the effect of different pre-training steps for the CliffWalking environment.

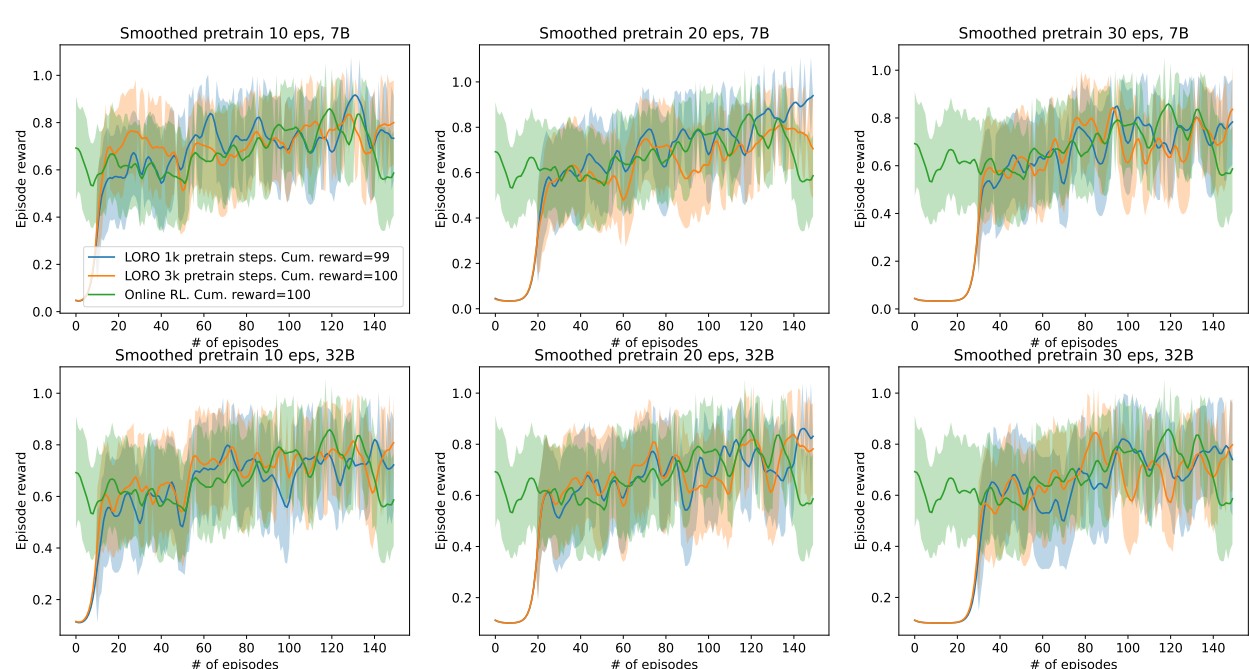

Figure 18: Comparing the effect of different pre-training steps for the FrozenLake environment.

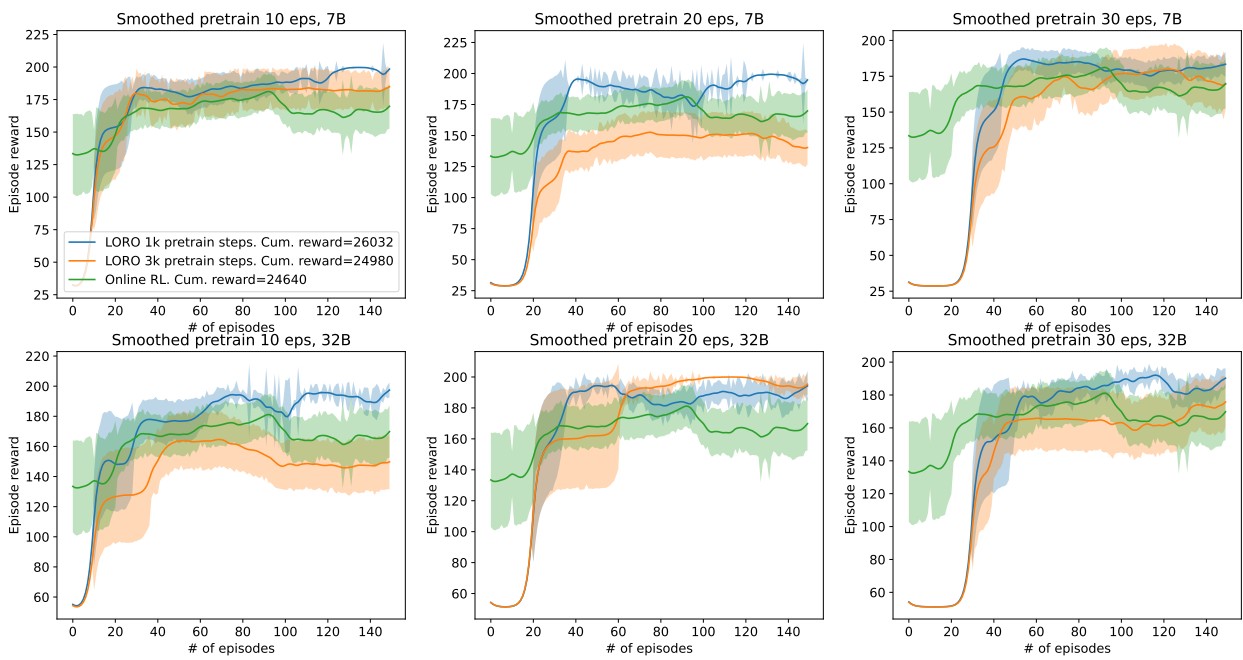

Figure 19: Comparing the effect of different pre-training steps for the CartPole environment.

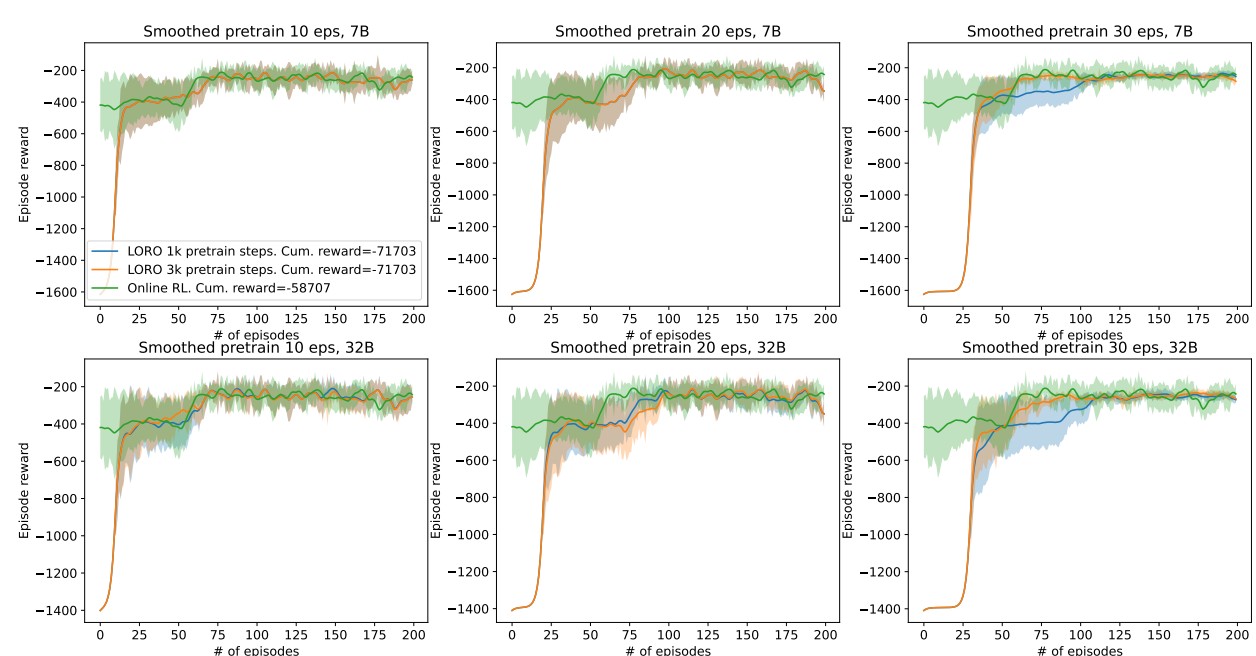

Figure 20: Comparing the effect of different pre-training steps for the Pendulum environment.

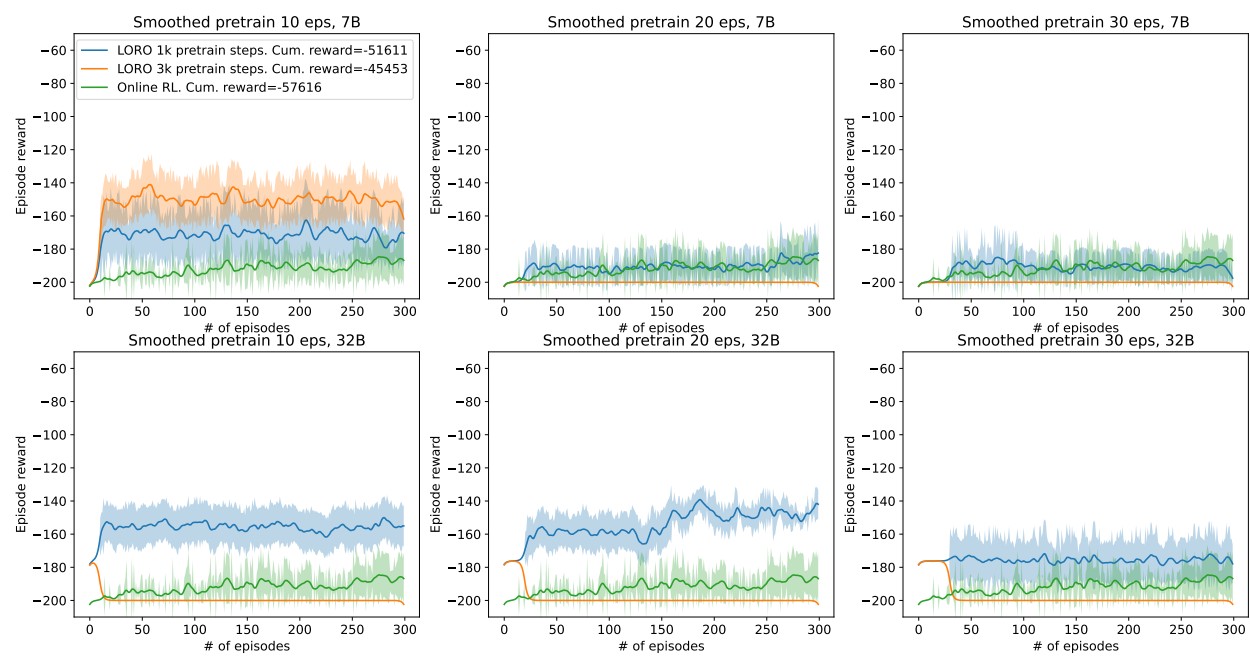

Figure 21: Comparing the effect of different pre-training steps for the MountainCar environment.

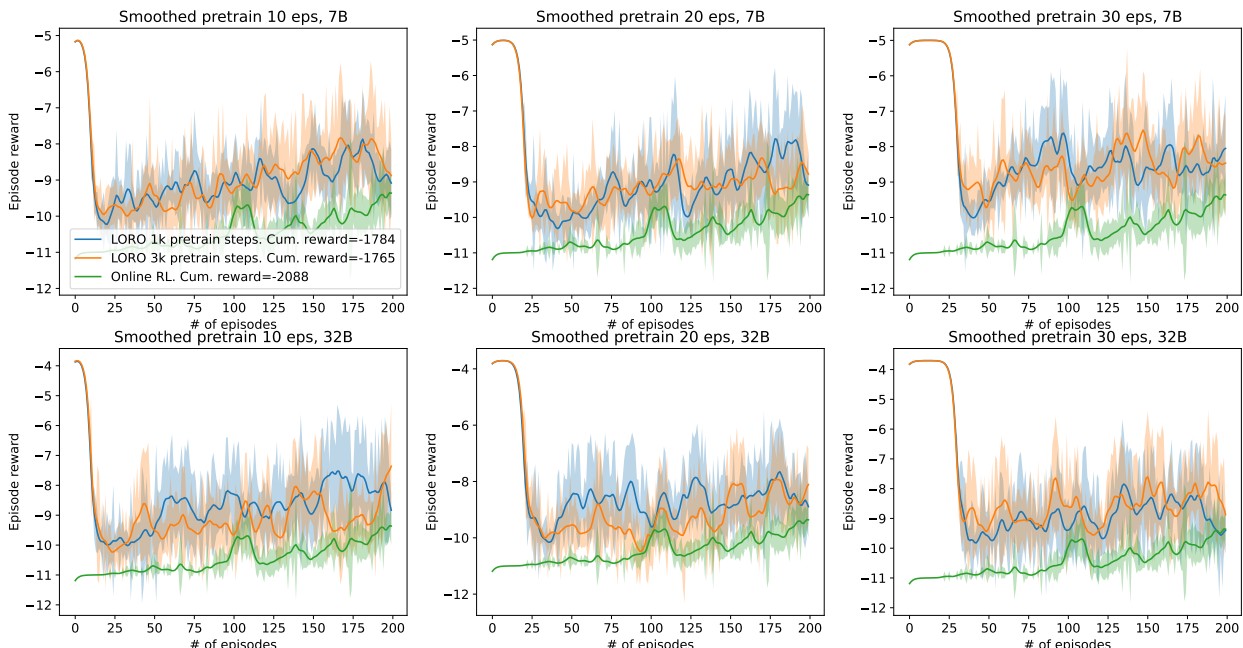

Figure 22: Comparing the effect of different pre-training steps for the Pong environment.

## E.4 EFFECTS OF THE AMOUNT OF LLM DATA

In this section, we evaluate the effect of the number of pre-training data on the cumulative reward of the policy. We evaluate this with different model sizes and pre-training steps on six OpenAI Gym environments and show the result in Figure 23, 24, 25, 26, 27, 28. Although there exist some differences in the cumulative reward, all baselines converge to a policy with similar performance in a relatively short amount of time. Hence, we observe no clear advantage of using a higher or lower amount of pre-training data to improve the decision-making quality of the LORO policy.

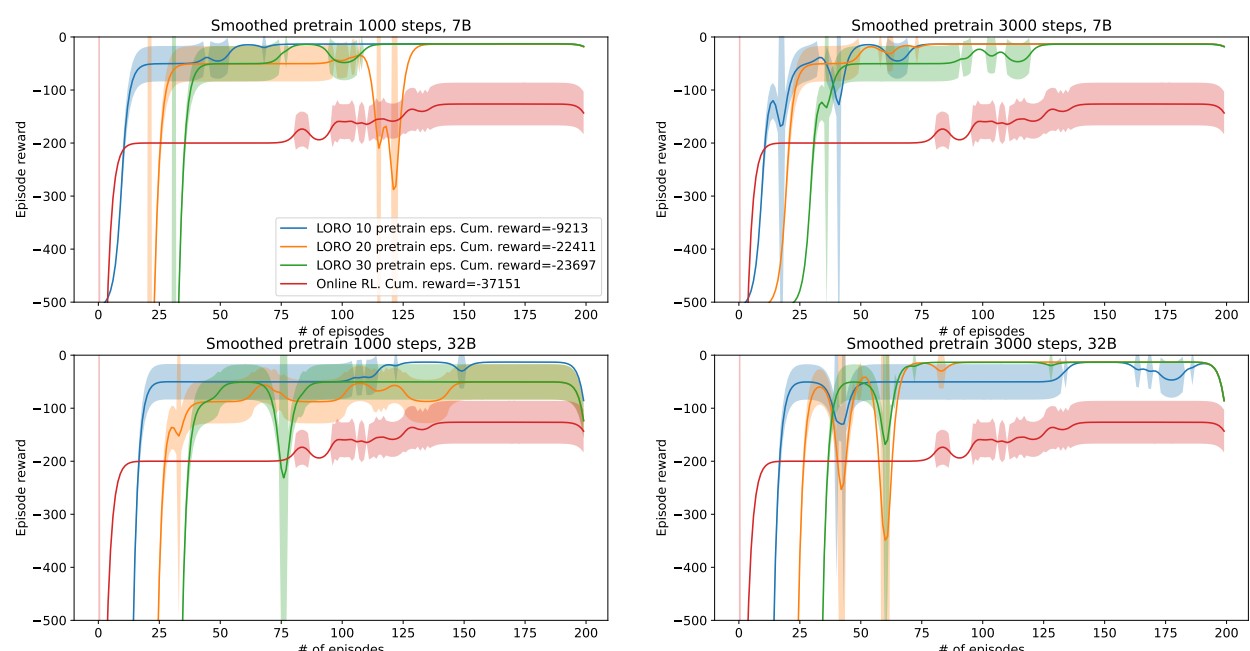

Figure 23: Comparing the effect of different amounts of pre-training data for the CliffWalking environment.

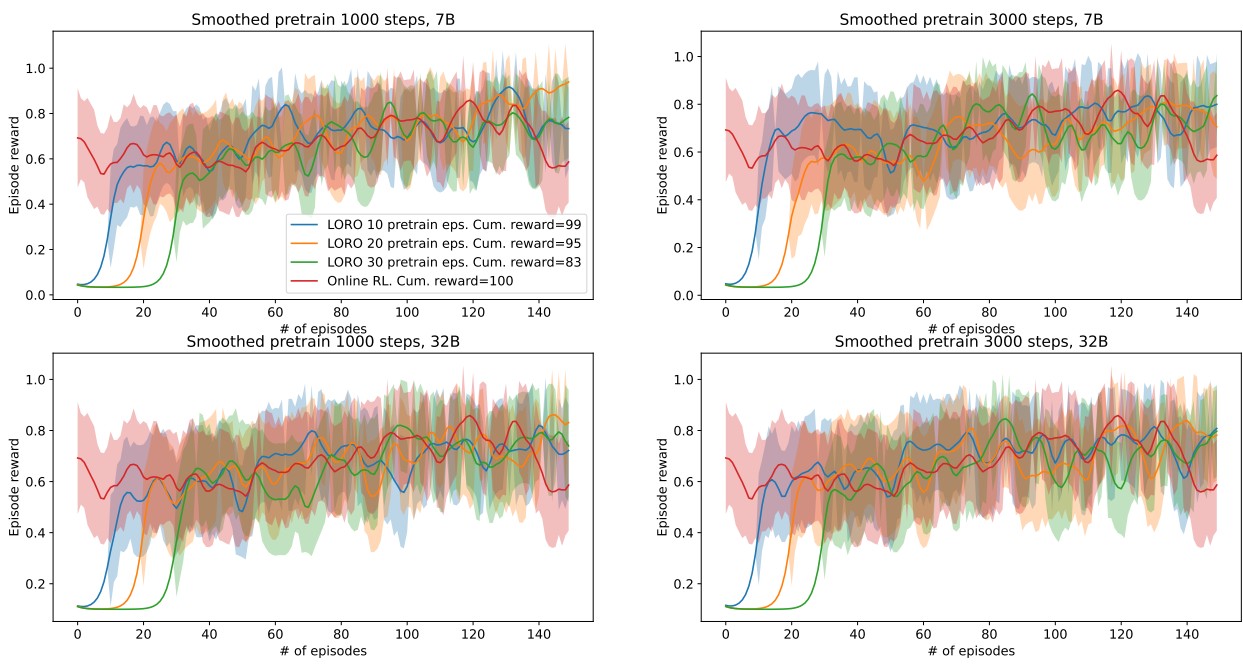

Figure 24: Comparing the effect of different amounts of pre-training data for the FrozenLake environment.

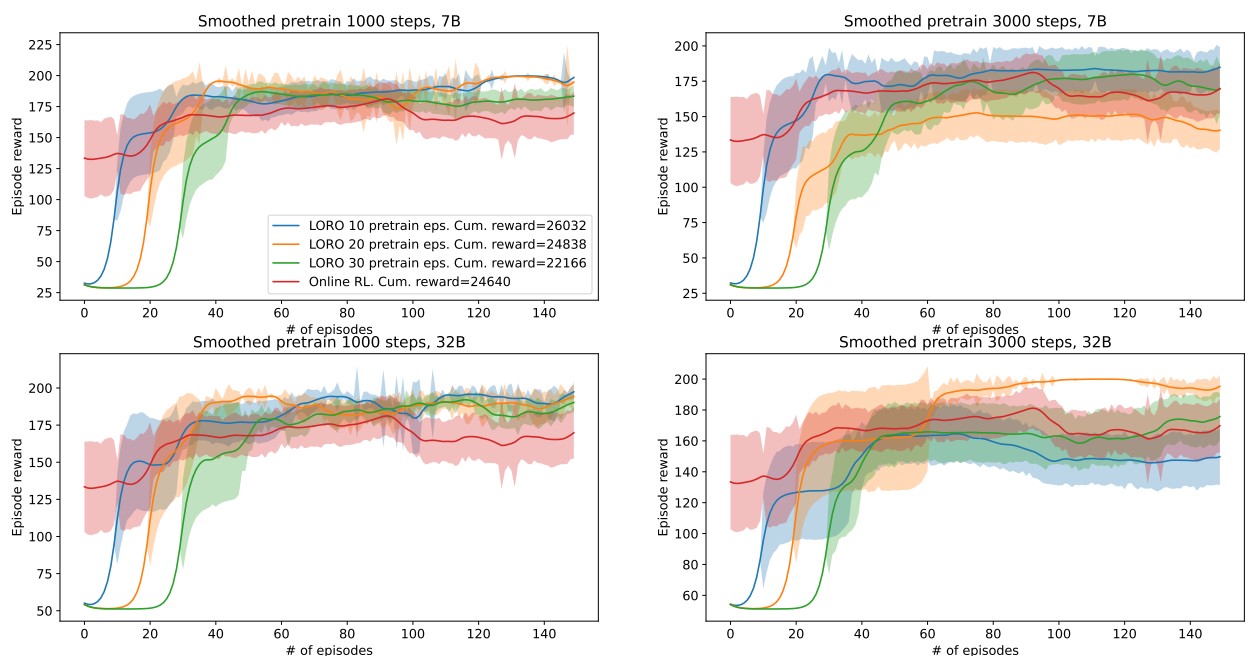

Figure 25: Comparing the effect of different amounts of pre-training data for the CartPole environment.

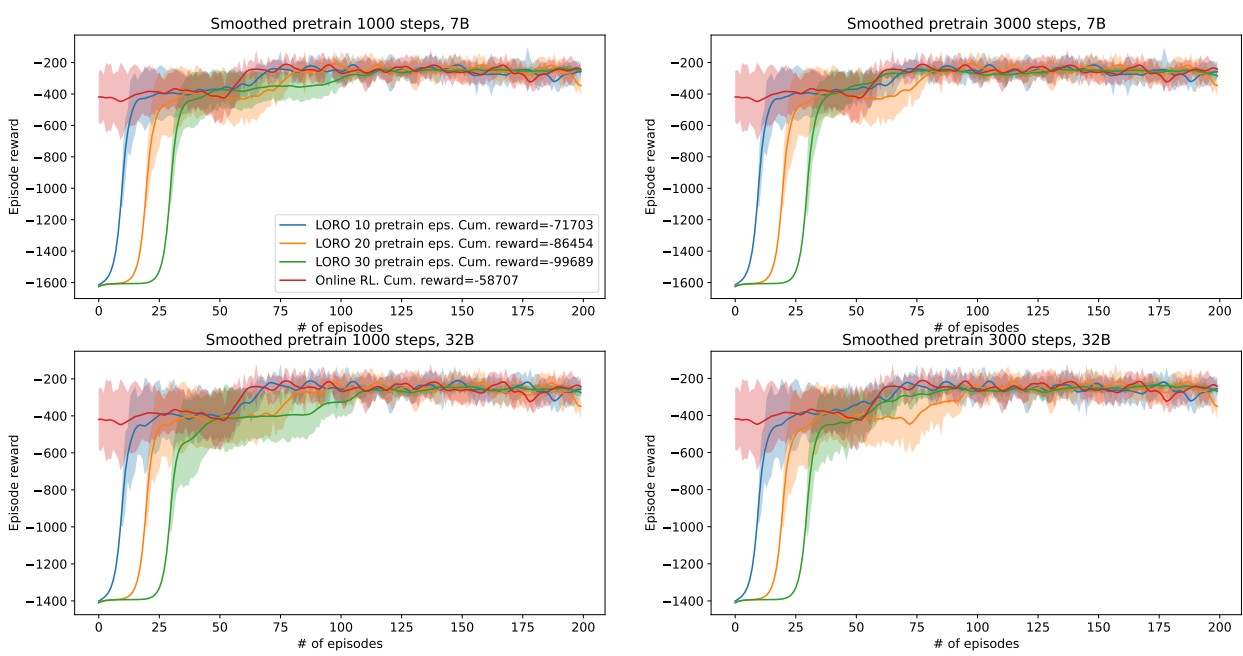

Figure 26: Comparing the effect of different amounts of pre-training data for the Pendulum environment.

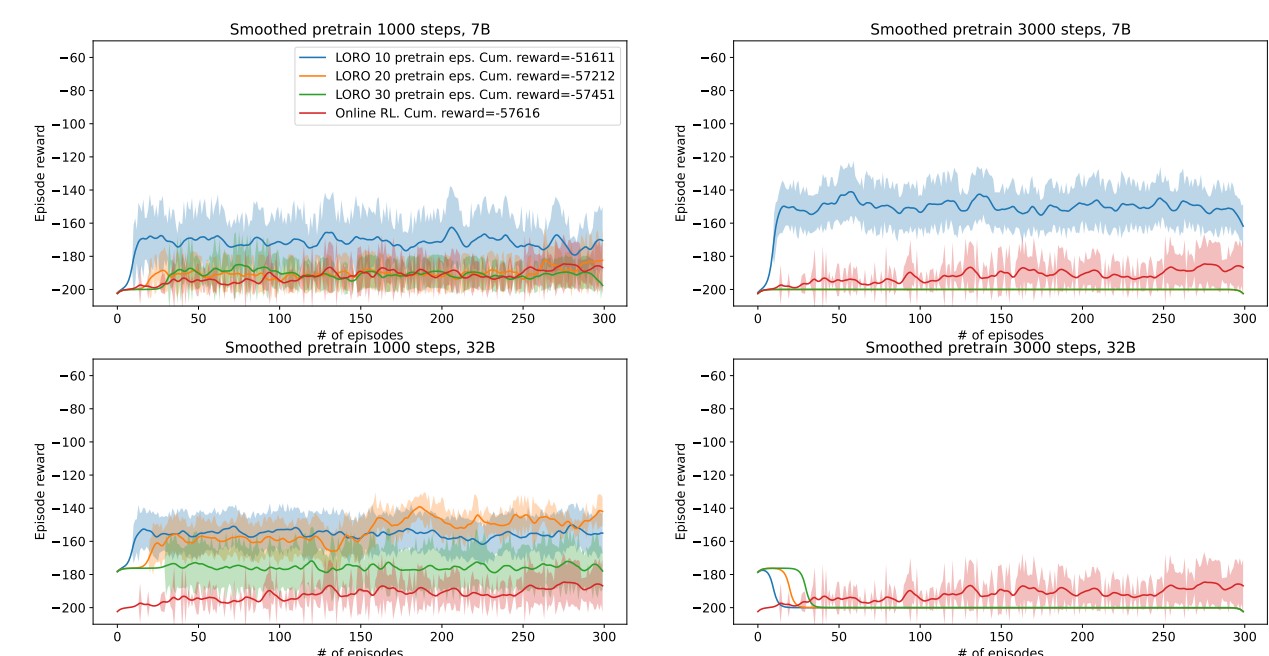

Figure 27: Comparing the effect of different amounts of pre-training data for the MountainCar environment.

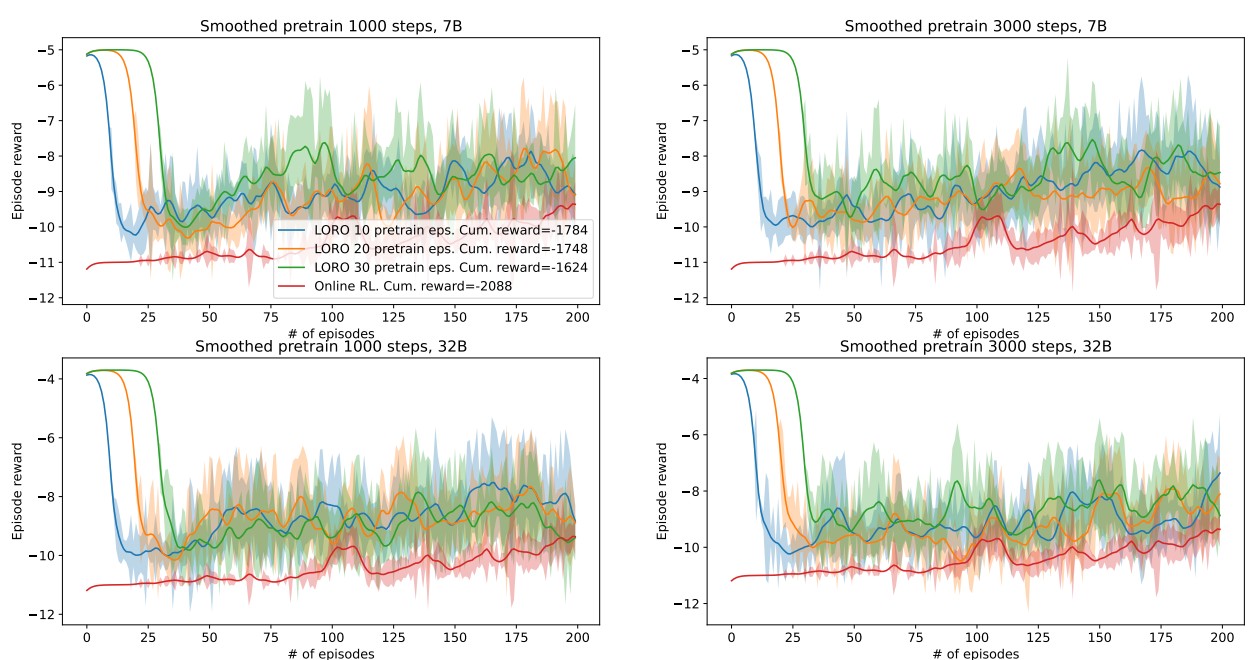

Figure 28: Comparing the effect of different amounts of pre-training data for the Pong environment.

## E.5 EFFECTS OF SFT AND LONG CoT

In this section, we evaluate the effect of SFT and Long CoT on the cumulative reward of the policy. We evaluate this with different pre-training data and pre-training steps on three OpenAI Gym environments and show the result in Figure 29, 30, 31. Overall, we observe no clear advantage of using SFT and Long CoT over vanilla CoT to improve the decision-making quality of the LORO policy.

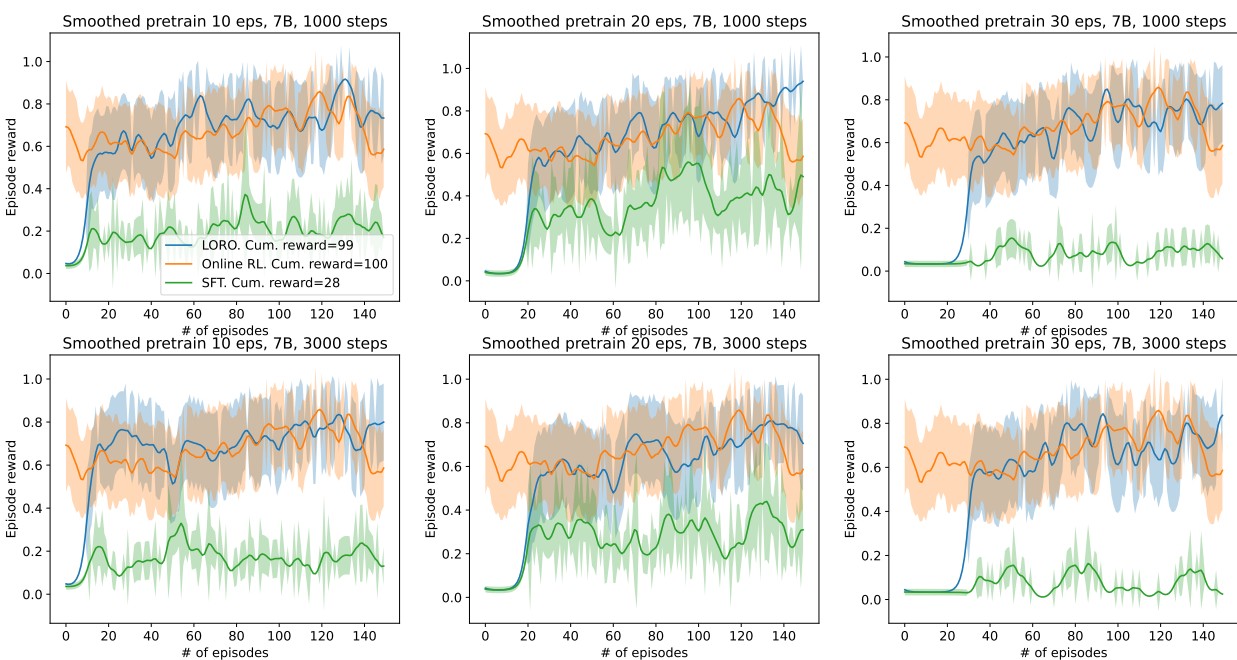

Figure 29: Comparing the effect of Long Chain-of-Thought and Supervised-Fine-Tuning for the FrozenLake environment.

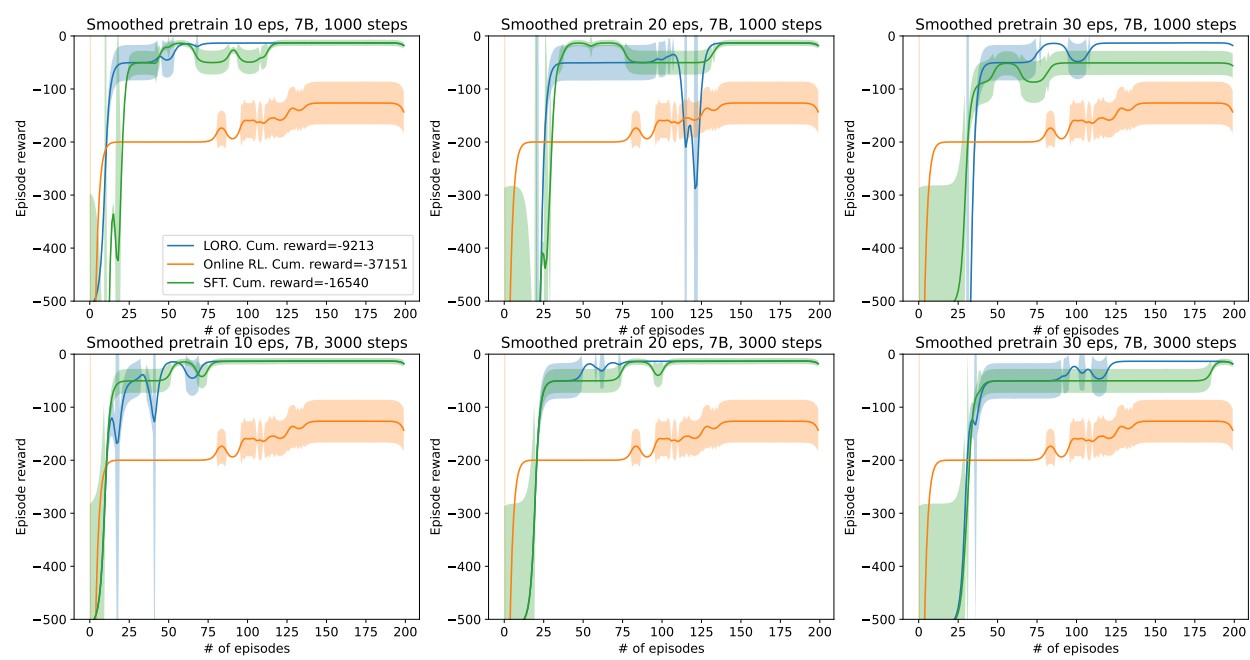

Figure 30: The effect of Supervised-Fine-Tuning for the CliffWalking environment.

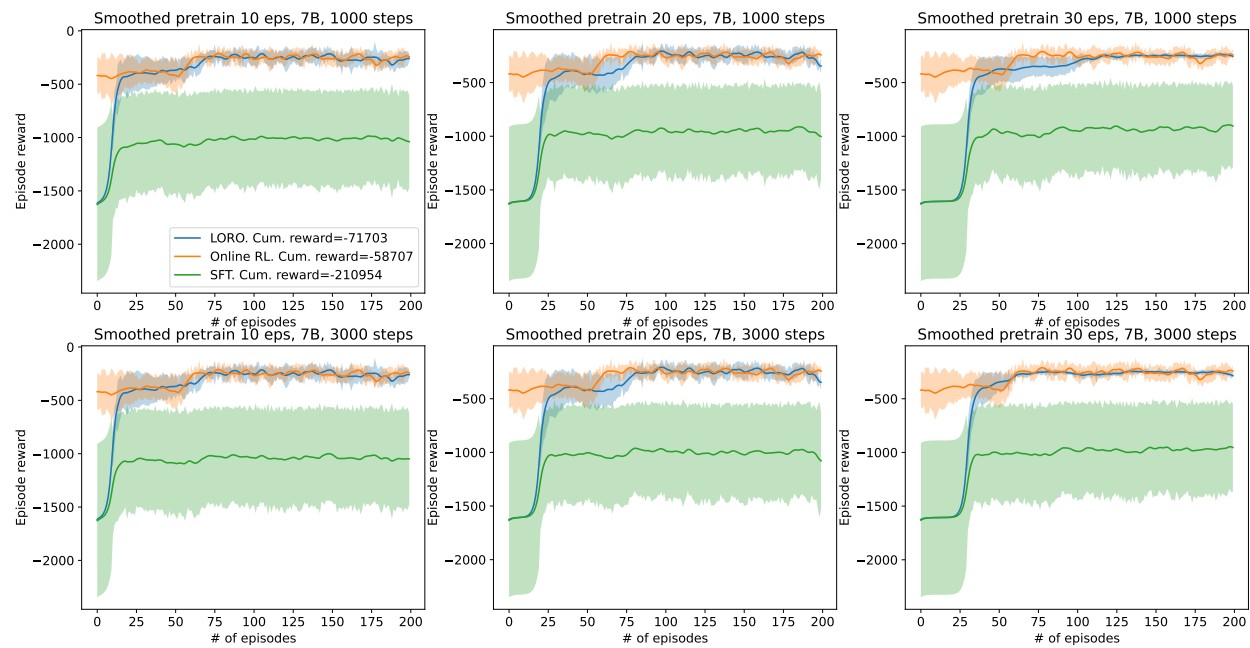

Figure 31: The effect of Supervised Fine-Tuning for the Pendulum environment.

## E.6 EFFECTS OF THE HISTORY SUMMARY

For the experiments above, we use an efficient environment history such as "The holes are in locations: X, Y, Z. You receive zero reward at locations: A, B, C, D".

For the experiment in Figure 32, we concatenate the observations of each state to the LLM's prompt, with a limited history length: "You visit location X and receive zero reward. You visit location Y and receive one reward. You visit ...".

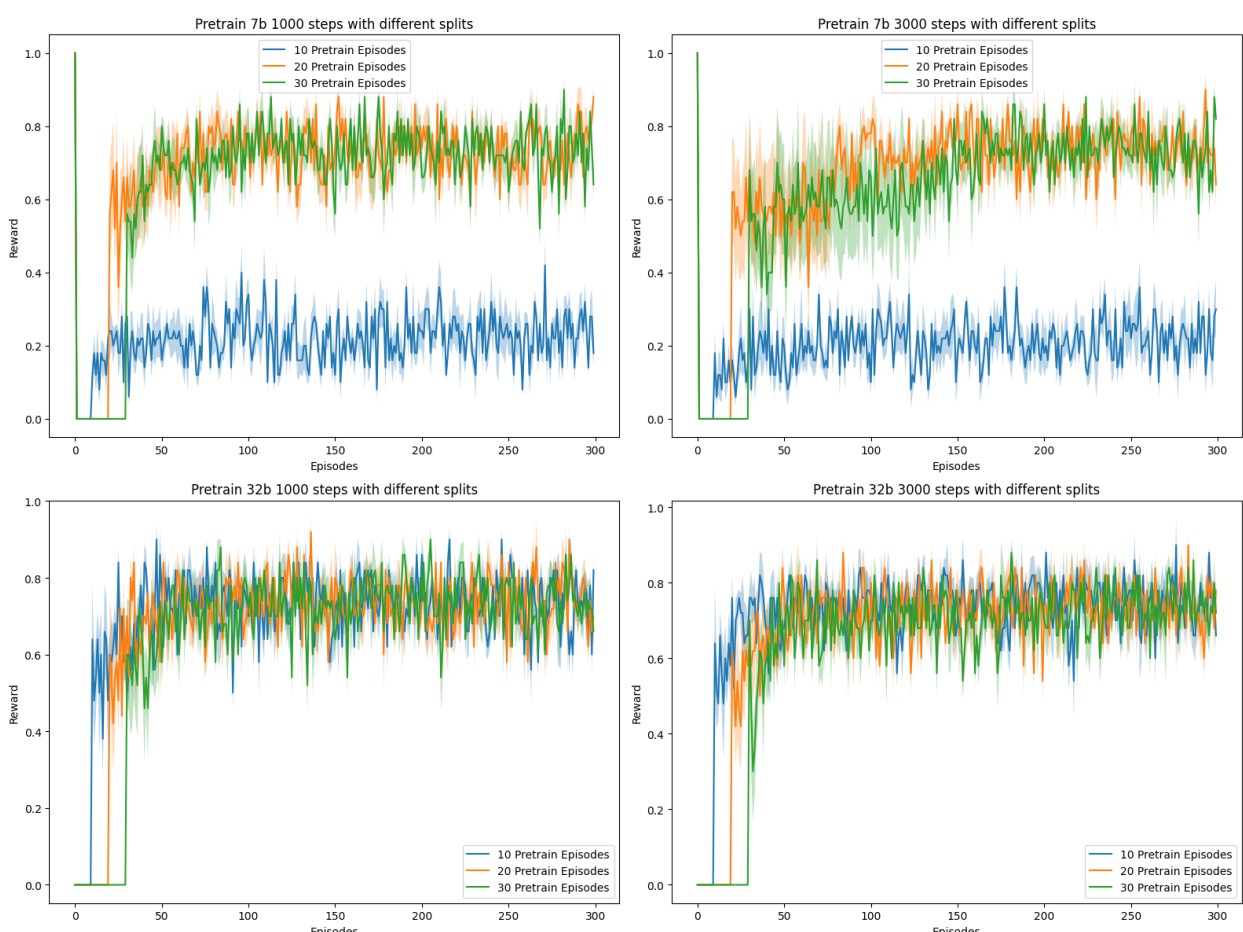

Figure 32: FrozenLake with ineffective environment history.

## F    WALL CLOCK TIME

Below, we show the wall-clock time for the LLM data collection phase in Table 5 and the fine-tune with RL phase in Table 6.

| Environment | Model | Time (h) | GPU |
|---|---|---|---|
| CliffWalking | Qwen-32b | 54.03 | H100 |
| | Qwen-7b | 19.96 | |
| CartPole | Qwen-32b | 15.66 | |
| | Qwen-7b | 8.27 | |
| FrozenLake | Qwen-32b | 2.63 | |
| | Qwen-7b | 0.82 | |
| MountainCar | Qwen-32b | 33.63 | |
| | Qwen-7b | 25.94 | |
| Pendulum | Qwen-32b | 59.68 | |
| | Qwen-7b | 28.5 | |
| RepresentedPong | Qwen-32b | 69.43 | |
| | Qwen-7b | 28.48 | |
| FrozenLake-LongCoT | DeepSeek-14b | 16.99 | A6000 |
| | DeepSeek-7b | 8.99 | |
| CliffWalking-SFT | Qwen-7b | 41 | H100 |
| FrozenLake-SFT | Qwen-7b | 0.73 | A6000 |
| Pendulum-SFT | Qwen-7b | 57.72 | |

Table 5: Wall-clock time for LLM data collection phase in different environments with different GPUs.

| Environment | Offline data | Offline data size (episode) | | |
|---|---|---|---|---|
| | | 10 | 20 | 30 |
| CliffWalking | Qwen-7b | 4.319 | 3.044 | 1.448 |
| | Qwen-32b | 2.149 | 1.249 | 0.788 |
| | Random | 4.661 | 2.138 | 1.922 |
| | Collected online | 5.131 | 2.392 | 2.045 |
| CartPole | Qwen-7b | 1.912 | 1.826 | 1.758 |
| | Qwen-32b | 1.432 | 1.354 | 1.275 |
| | Random | 2.707 | 2.705 | 2.718 |
| | Collected online | 2.992 | 2.993 | 3.005 |
| FrozenLake | Qwen-7b | 1.896 | 1.797 | 1.74 |
| | Qwen-32b | 1.397 | 1.318 | 1.254 |
| | Random | 2.746 | 2.741 | 2.757 |
| | Collected online | 3.069 | 3.068 | 3.059 |
| MountainCar | Qwen-7b | 2.177 | 2.009 | 1.112 |
| | Qwen-32b | 1.608 | 1.446 | 0.8 |
| | Random | 3.155 | 3.11 | 4.808 |
| | Collected online | 3.513 | 3.253 | 3.21 |
| Pendulum | Qwen-7b | 3.803 | 3.647 | 2.534 |
| | Qwen-32b | 2.806 | 2.658 | 1.818 |
| | Random | 5.552 | 5.227 | 3.979 |
| | Collected online | 6.159 | 4.552 | 4.358 |
| RepresentedPong | Qwen-7b | 3.639 | 3.26 | 2.69 |
| | Qwen-32b | 2.386 | 2.406 | 1.99 |
| | Random | 4.295 | 3.253 | 2.319 |
| | Collected online | 4.63 | 3.444 | 2.434 |

Table 6: Wall-clock time for the fine-tune with RL phase with an H100 GPU. The only exception is the MountainCar with Offline data size of 30, which was timed using an A6000 GPU.

# G  LLM SETUP

We designed the prompt to choose an action from a list of integers starting from one, since we observed that LLM is more biased toward action zero. After the LLM chooses an action, we extract it by getting the last number returned by the LLM. This design was inherited from Pandey (2024), which can be improved since we observe a number of extraction failures from our experiments.

We observe that the vanilla design of LLM, where we ask it only to return the chosen action, performs poorly. Similarly, we implemented and tested the Majority Voting and Best-of-N test-time-scaling methods, but they both perform poorly without CoT.

For all experiments, we limit the generating token to be less than 2000 , top-p 0.6, top-k 0, temperature 0.9. In the SFT experiment, the LLM was trained with LoRA Hu et al. (2021) with rank 8, alpha 16, dropout 0.05, batch size 1, and using 8-bit quantization.

# H  PROMPTS

This section provides a collection of Chain-of-Thought (CoT) prompting examples used across multiple decision-making environments. Each task is presented with a detailed system prompt describing the environment dynamics and action space, followed by a user query that reflects a specific game state. The responses are generated by `Qwen2.5-7B-Instruct`, illustrating its step-by-step reasoning capabilities.

## H.1  CHAIN-OF-THOUGH PROMPTS

### H.1.1  TASK 1: CLIFF WALKING

---

**A Sample of Cliff Walking**

**► System:**
You are an expert-level game player. Cliff walking is a task in which you control a player navigating a '4x12' grid world. The ('x', 'y') coordinate indicates the position at row 'x' and column 'y'. The player needs to find a goal location while avoiding cliffs . The player can choose from 4 actions: move up, move right, move down, or move left. If the player takes an action at ('x', 'y'), he tries to move to ('a', 'b'). Rules:

- 1. If ('a', 'b') is a cliff, the player incurs a large penalty of -100, and is reset to the starting position.
- 2. If ('a', 'b') is safe or towards the grid boundary, results in a small penalty of -1. If ('a', 'b') is outside the grid's boundaries, it does not change position but still receive the -1 penalty.
- 3. The game ends when the ('a', 'b') is the goal or 200 actions are performed. The goal is to navigate from the starting point to an target , while avoiding the cliff, in as few steps as possible.

Type a number to indicate the action. For current position ('x', 'y'), the action means the player try to step into the next position. Type '1' to move up, which means trying to step into ('x-1', 'y'), '2' to move right, which means ('x', 'y+1'), '3' to move down, which means ('x+1', 'y'), or '4' to move left, which means ('x', 'y-1'). Ensure you only provide the action number from the valid action list, i.e., [1, 2, 3, 4]. Environment history: Cliff: Reward -100 at locations: (3, 0). Reward -1 at locations: (2, 0), (3, 0), (1, 0). Previous location: (1, 0), previous action: 1, previous reward: -1. Return the action at the end of your answer without the target's location.

**► User:**
You are at location (2, 0) in the grid world.
Think step by step.

**► Qwen2.5-7B-Instruct:**
1

---

### H.1.2 TASK 2: CARTPOLE

---

**A Sample of CartPole**

▶ **System:**
You are an expert-level game player. In the CartPole game, you control a cart that moves along a horizontal track. There is a pole standing upright on the cart. The goal of the game is to keep the pole balanced upright by moving the cart left or right. The game ends if the pole tilts too far from the vertical position or if the cart moves too far from the center of the track. The longer you can keep the pole balanced, the higher your score.Note that when the Cart Position is out of the (-2.4, 2.4) zone or the Pole Angle is out of the zone (-.2095, .2095), the round ends and the game is lost. The goal is to keep the pole balanced upright for as long as possible.
Type '1' to push the cart to the left or '2' to push the cart to the right. Ensure you only provide the action number from the valid action list, i.e., [1, 2].

▶ **User:**
The cart is positioned at 0.006, with a velocity of 0.04 towards the right. The pole is tilted at 0.02 radians, rotating at 0.02 radians per second towards the right.
Think step by step.

▶ **Qwen2.5-7B-Instruct:**
Given the current state of the CartPole game:

- Cart Position: 0.006 (slightly to the right of the center)
- Cart Velocity: 0.04 (moving slightly to the right)
- Pole Angle: 0.02 radians (slightly tilted to the right)
- Pole Angular Velocity: 0.02 radians/second (tilting to the right)

Let's analyze the situation step by step:

1. **Cart Position**: The cart is very close to the center, at 0.006. This is within the safe range, so we don't need to move it significantly.
2. **Cart Velocity**: The cart is moving slightly to the right, which is good because it helps counteract the tilt of the pole.
3. **Pole Angle**: The pole is tilted 0.02 radians to the right. This is a small tilt, but it indicates that the pole is starting to lean. We need to act quickly to prevent it from falling over.
4. **Pole Angular Velocity**: The pole is rotating at 0.02 radians/second to the right, which means it is accelerating its tilt. This suggests that the cart's movement is not sufficient to stabilize the pole.

Given these observations, we should try to slow down the pole's tilt by moving the cart to the left. This will help counteract the rightward tilt of the pole.

**Action: 1 (Push the cart to the left)**

This action will help to stabilize the pole by moving the center of mass back towards the center, which should reduce the angular velocity of the pole.

---

### H.1.3 TASK 3: MOUNTAIN CAR

---

**A Sample of Mauntain Car**

▶ **System:**
You are an expert-level game player. In the Mountain Car game, you control a car placed stochastically at the bottom of a sinusoidal valley. The only possible actions are the accelerations that can be applied to the car in either direction. The goal of the game is to strategically accelerate the car to reach the goal state on top of the right hill as quickly as possible. The episode ends if either the car reaches the goal position on top of the right hill or the length of the episode is 200. The goal is to reach the flag placed on top of the right hill as quickly as possible.
Type '1' to accelerate to the left, '2' to not accelerate, or '3' to accelerate to the right.Ensure you only provide the action number from the valid action list, i.e., [1, 2, 3].

---

▶ **User:**
The car is positioned at -0.541, with a velocity of 0.000 towards the left.
Think step by step.

▶ **Qwen2.5-7B-Instruct:**
Given the car's position at -0.541 and a velocity of 0.000 towards the left, the car is slightly to the left of the center of the valley. Since the car is not moving yet and is close to the middle, accelerating to the right (action 3) might help gain some speed towards the right hill without overshooting too much.

Action: 3

### H.1.4   TASK 4: FROZENLAKE

**A Sample of FrozenLake (CoT)**

▶ **System:**
You are an expert-level game player. In the FrozenLake game, the player starts at the start position of the grid and tries to reach the goal position . There are holes which the player must avoid. The frozen lake is slippery, meaning that the player might not always move in the intended direction. The game ends when the player reaches the goal or falls into a hole. The goal is to navigate across the frozen lake and reach the goal position without falling into any holes. For current position ('x', 'y'), the action means the player try to step into the next position. The possible actions are:

- 1: Move left, which means ('x', 'y-1'),
- 2: Move down, which means ('x+1', 'y'),
- 3: Move right, which means ('x', 'y+1'),
- 4: Move up, which means trying to step into ('x-1', 'y').

Ensure you only provide the action number from the valid action list, i.e., [1, 2, 3, 4]. Do not return the target's coordination. Environment history: Reward 0.0 at locations: (0, 0). Previous location: (0, 0), previous action: 1, previous reward: 0.0. Return the action at the end of your answer without the target's location.

▶ **User:**
You are at row 0, column 0.
Think step by step.

▶ **Qwen2.5-7B-Instruct:**
Since the current position is (0, 0) and the previous action was moving left (which is not possible as it would take you out of bounds), the only feasible action to consider is moving down to (0, 1). Therefore, the action is: **2**.

**A Sample of FrozenLake (Long CoT)**

▶ **System:**
You are an expert-level game player. In the FrozenLake game, the player starts at the start position of the grid and tries to reach the goal position . There are holes which the player must avoid. The frozen lake is slippery, meaning that the player might not always move in the intended direction. The game ends when the player reaches the goal or falls into a hole. The goal is to navigate across the frozen lake and reach the goal position without falling into any holes. For current position ('x', 'y'), the action means the player try to step into the next position. The possible actions are:

- 1: Move left, which means ('x', 'y-1'),
- 2: Move down, which means ('x+1', 'y'),
- 3: Move right, which means ('x', 'y+1'),
- 4: Move up, which means trying to step into ('x-1', 'y').

Ensure you only provide the action number from the valid action list, i.e., [1, 2, 3, 4]. Do not return the target's coordination. Environment history: Reward 0.0 at locations: (0, 0). Previous location: (0, 0), previous action: 1, previous reward: 0.0. Return the action at the end of your answer without the target's location.

► **User:**
The current position of the You are at row 0, column 0.
Think step by step.

► **DeepSeek-R1-Distill-Qwen-7B:**

Okay, so I'm trying to figure out where to move in this FrozenLake game. I'm at position (0,0), which is the top-left corner of the grid. The goal is to reach the goal position, which I think is at the bottom-right corner, maybe (4,4) since it's a 5x5 grid. But I'm not entirely sure, so I'll assume that's the case.
First, I need to consider the possible actions. The actions are numbered 1 to 4, corresponding to moving left, down, right, and up respectively. Each action has a direction, but because the ice is slippery, I might not end up where I intended. So, I have to plan carefully to avoid falling into holes.

Let me think about each action:

- Action 1: Move left. But wait, I'm already at column 0, which is the leftmost column. So moving left from here would take me off the grid, which is probably a hole. So I shouldn't choose action 1.

- Action 2: Move down. That would take me to (1,0). I need to check if that's a hole. From the grid, I remember that some cells are safe and some are holes. Let me visualize the grid:

  – Row 0: S F F F F
  – Row 1: F F H F F
  – Row 2: F H F F F
  – Row 3: F F F H F
  – Row 4: F F F F G

  Wait, I'm not sure if that's accurate, but I think row 1, column 2 is a hole (H), and row 2, column 1 is another hole. So moving down to (1,0) is safe because row 1, column 0 is F (safe). So action 2 seems safe.

- Action 3: Move right. That would take me to (0,1). I think that's safe too because row 0, column 1 is F.

- Action 4: Move up. But I'm at row 0, so moving up would take me off the grid, which is a hole. So action 4 is out.

So, from (0,0), the valid actions are 2 and 3. I need to choose between them. Let's think about the long-term implications.
If I go down to (1,0), then from there, possible actions are:

- Move left: off grid (hole)

- Move down: (2,0) which is safe

- Move right: (1,1) which is safe

- Move up: back to (0,0)

If I go right to (0,1), then from there:

- Move left: back to (0,0)

- Move down: (1,1) which is safe

- Move right: (0,2) which is safe

- Move up: off grid (hole)

Hmm, both paths seem viable. Maybe I should go right to (0,1) because moving down from there might lead me closer to the goal without getting stuck. Alternatively, going down might allow me to move right or left without immediate issues.

Wait, but if I go down to (1,0), then moving right to (1,1) is safe, and from there, moving down again to (2,1) is a hole because row 2, column 1 is H. Oh no, that's a problem. So from (1,0), if I move right to (1,1), then down to (2,1) is a hole. So that's bad.
Alternatively, from (1,0), if I move down to (2,0), which is safe, then from there, moving right to (2,1) is a hole. So again, that's a problem.

Wait, maybe I'm misremembering the grid. Let me try to recall the layout. I think row 1, column 2 is H, and row 2, column 1 is H. So, from (1,0), moving right goes to (1,1), which is safe, but then moving down from there would go to (2,1), which is H. So that's a problem.

Alternatively, from (1,0), moving down to (2,0) is safe, but then moving right to (2,1) is H. So either way, moving down first might lead to a hole in the next step.

So maybe going right to (0,1) is better. From (0,1), moving down to (1,1) is safe, and then from there, moving right to (1,2) is H, which is bad. Or moving right to (0,2), which is safe, then from there, moving down to (1,2) is H. So again, same issue.

Wait, maybe I'm overcomplicating this. Perhaps I should try to find a path that avoids the holes. Let's think about possible paths.

From (0,0), go right to (0,1). From there, go down to (1,1). From (1,1), go right to (1,2), but that's H. So that's bad. Alternatively, from (1,1), go right to (1,2) is H, so maybe go down to (2,1), which is H. So that's no good. Alternatively, from (0,1), go down to (1,1), then from there, go right to (1,2) is H, or left to (1,0), which is back where I came from. Hmm.

Wait, maybe from (0,1), go right to (0,2). From (0,2), go down to (1,2) is H. So that's bad.

Alternatively, from (0,1), go down to (1,1), then from there, go right to (1,2) is H, or left to (1,0), which is safe. So from (1,1), moving left to (1,0) is safe, but then from (1,0), moving down to (2,0) is safe, and from there, moving right to (2,1) is H. So that's a problem.

This is getting complicated. Maybe I should try a different approach. Let's think about the possible paths step by step.

From (0,0), options are 2 (down) and 3 (right).

If I go down to (1,0):

- From (1,0), possible actions: down to (2,0), right to (1,1)

- If I go down to (2,0):

  - From (2,0), possible actions: right to (2,1) is H, so that's bad. So maybe go right to (2,1) is H, so can't go there. So from (2,0), the only safe move is right to (2,1) is H, so maybe move up to (1,0), but that's where I came from. Or move right to (2,1) is H, so that's not allowed. So from (2,0), the only safe move is to go right to (2,1), but that's H, so that's bad. Alternatively, move left to (2,-1), which is off grid, so that's a hole. So from (2,0), I can't move anywhere without falling into a hole. So that's a dead end.

So going down to (1,0) leads to a dead end because from (2,0), I can't move anywhere safely.

So, maybe I should go right to (0,1) instead.

From (0,1):

- Possible actions: down to (1,1), right to (0,2)

- If I go down to (1,1):

  - From (1,1), possible actions: down to (2,1) is H, right to (1,2) is H, left to (1,0), up to (0,1)

  - So, from (1,1), the only safe move is left to (1,0) or up to (0,1). If I go left to (1,0), then from there, I can go down to (2,0), but from (2,0), as before, it's a dead end. Alternatively, from (1,1), go up to (0,1), which is where I came from, so that's not helpful.

- If I go right to (0,2):

  - From (0,2), possible actions: down to (1,2) is H, right to (0,3), left to (0,1)

  - So, from (0,2), the safe moves are right to (0,3) or left to (0,1). If I go right to (0,3), then from there, I can go down to (1,3), which is safe, and so on.

So, from (0,1), going right to (0,2) seems better because it allows me to move further right without hitting a hole immediately.

So, let's try that path:
**Action: 1**

### H.1.5 TASK 5: PONG

---

**A Sample of Pong**

▶ **System:**
You are an expert-level game player. In the Pong game, you play the ball with your opponent, each player rallys the ball by moving the paddles on the playfield. Paddles move only vertically on the playfield. A player scores one point when the opponent hits the ball out of bounds or misses a hit. The first player to score 21 points wins the game. The number of frameskip is set to 4. The goal is to knock out your opponent. Type 1 for NOOP (no operation), 3 to move up, 4 to move down. Ensure you only provide the action number from the valid action list, i.e., [1, 3, 4].

▶ **User:**
The origin (0,0) is in the top left corner. You are at position (188, 78), your opponent is at position (64, 8), the ball is at (0, 0), the ball velocity is (0, -60). Your opponent's score is 0, your score is 0.
Think step by step.

▶ **Qwen2.5-7B-Instruct:**
Given the current state of the game:

- Your paddle is at position (188, 78).
- Your opponent's paddle is at position (64, 8).
- The ball is at position (0, 0) and is moving upwards with a velocity of (0, -60).

**Step-by-step reasoning:**

- The ball is at the top of the playfield and is moving downwards. It will be near the middle of the playfield soon.
- With a frameskip of 4, the opponent will not react until the ball has moved significantly downward.
- Since the ball is about to come towards you, you should prepare to move your paddle to intercept the ball.

**Action Plan:**

- Move your paddle up to intercept the ball as it comes towards you. This ensures that you can return the ball effectively.

**Action: 3** (move up)
By moving up, you position yourself to intercept the ball as it comes towards the center of the playfield.

---

### H.1.6 TASK 6: PENDULUM

**A Sample of Pendulum**

▶ **System:**
You are an expert-level game player. In the Pendulum game, you control a pendulum attached to a fixed pivot point. The goal is to apply torques to swing the pendulum upright and keep it balanced. The game ends if the pendulum cannot be stabilized within the given time limit. The closer the pendulum is to the upright position, the higher your score. The goal is to swing the pendulum upright and keep it balanced. Provide a torque value (e.g., a float between -2.0 and 2.0) to control the pendulum's movement. Return the torque value enclosed in < and >, e.g., <1.5>.

▶ **User:**
The pendulum is at an angle of -2.690 radians from the vertical (zero when upright), rotating at 0.34 radians per second in the clockwise direction.
Think step by step.

▶ **Qwen2.5-7B-Instruct:**
To determine the appropriate torque to apply, we need to consider the current state of the pendulum: its angle and angular velocity.

- **Angle Analysis**: The pendulum is currently at an angle of -2.690 radians from the vertical. This means it is hanging to the left of the vertical position. To bring it back towards the vertical, we need to apply a

positive torque (since positive torque will tend to rotate the pendulum clockwise, which is the direction needed to move it towards the vertical).

- **Angular Velocity Analysis**: The pendulum is rotating at 0.34 radians per second in the clockwise direction. Since the pendulum is already moving in the direction we want (towards the vertical), we should continue to apply a positive torque to help it swing more quickly towards the vertical position.

Given that the pendulum is already moving in the correct direction but not fast enough, applying a moderate positive torque should help it swing faster and reach the vertical position more efficiently.

A good starting torque value could be around 1.0, as this provides a reasonable amount of force without being too aggressive and risking overshooting the vertical position.

Therefore, the torque value I suggest is: **<1.0>**

