# OpenReview forum: "Sample-efficient Reinforcement Learning by Warm-starting with LLMs"
_ICLR.cc/2026/Conference — Submitted to ICLR 2026_

### Official Review · Reviewer_ZgGw · 2025-10-26

**Soundness:** 2
**Presentation:** 2
**Contribution:** 2
**Rating:** 4
**Confidence:** 4

**Summary:**

This paper introduces an algorithm called LLM Offline, RL Online (LORO) which is designed to make reinforcement learning faster and more data-efficient via pretraining with high quality offline data. The core idea is to "warm-start" the learning agent. Instead of starting from scratch with no knowledge, the method first uses a large language model to act as an initial policy and play the game for a few episodes. The idea is that this LLM will generate a small, high-quality offline dataset of experiences. A standard reinforcement learning algorithm is then pre-trained on this LLM-generated data before it starts interacting with the environment to learn and fine-tune its policy online. The authors argue this works because the language model's data, while not perfect, provides better coverage of important state-action pairs than random exploration. Their experiments on several game environments show that the LORO approach can speed up learning compared to starting with random exploration and online RL.

**Strengths:**

- The paper is highly reproducible and code is included which is a big plus. It includes detailed hyperparameter values, clear details of the LLM variations used, prompt formats, rollout budgets, and pretraining steps, along with a working anonymous code link.
- The core idea is novel, interesting, and exciting and provides a different modality for pretraining and general training of RL agents that could be quite useful in the future for reducing sample complexity of RL algorithms.
- The experiments are carefully designed to show what actually drives improvement. The paper compares pretraining on LLM data with mixing data directly, tests different data sources such as LLM, online, and random, and uses multiple RL algorithms showing that in off-policy settings the LORO idea can help.

**Weaknesses:**

- The writing quality throughout the paper is a concern. There are frequent grammatical mistakes, awkward sentence constructions, and unclear phrasing that collectively make the paper difficult to read through cleanly. I initially began noting specific instances to include in this review, but by the end of the related work section I had already identified at least five clear grammatical or syntactic errors and decided to stop recording them. These issues give the paper a rushed and unpolished feel. I recommend a thorough language and structure revision.

- The proposed approach relies on an external framework that provides a text-based wrapper enabling the LLM to interact with classic RL environments. Although this design choice is quite interesting theoretically, it introduces a strong dependency on environment-specific engineering. If I understand the prior work correctly, the wrapper translates numerical observations into text descriptions and maps LLM-generated text back into executable actions, which encodes task-specific structure and domain knowledge. This dependence limits the scalability and generality of the approach, as the method remains tied to environments for which wrappers already exist or can be easily synthesized manually. As a result, it is unclear how easily the framework could extend to new or more complex domains without substantial additional interface design. Likewise the question (which maybe is more appropriate for prior works) is how susceptible is this wrapper framework to reward hacking and prompt engineering failures. If the wrapper fails to work properly the ideas presented in Assumption 1 fall apart. Furthermore, the paper does not justify why this level of manual engineering would be better spent on using LLM-generated policies rather than equally direct approaches such as incorporating physics-informed priors, structured exploration strategies, or hand-coded heuristic controllers, which could provide similar sample efficiency benefits when a priori information about the environment’s structure is available.

- The paper's central methodological benefit hinges on Assumption 1, which claims that the LLM policy generates pretraining trajectories that sufficiently cover the state-action space that would be visited by an optimal policy. This assumption is foundational to the algorithm's warm start success, yet it lacks strong theoretical or mechanistic justification. The paper does not provide a clear reason why an LLM's text based reasoning should align with the optimal visitation distribution of an arbitrary MDP, especially in complex environments where optimal solutions are non intuitive. The only validation for this critical assumption is empirical, and this evidence is limited to simple gridworlds (CliffWalking, FrozenLake) where optimal paths are easily inferred from textual descriptions. The scalability of Assumption 1 to high dimensional, continuous, or partially observable domains remains unproven. I skimmed the appendix as well to try and better understand the assumption, but the empirical data in Appendix C (Table 3) is inconclusive and, in some cases, contradictory. For example, in the CliffWalking environment, the online collected data achieves a superior (lower) surrogate transfer coefficient than the LLM policies. The paper's post hoc dismissal of this data by suggesting the final policy used for evaluation was non optimal undermines the validity of its own verification protocol. This leaves the core premise that the LLM policy inherently provides high quality, optimal state covering data unconvincingly supported.

- Comparative performance is a weakness in this work. Across the six OpenAI Gym-style environments evaluated for sample efficiency, LORO only outperforms a simple online RL algorithm in three of the six cases. The only setting where LORO substantially outperforms prior work is in the CliffWalking environment, which could reasonably be classified as an edge case characterized by extreme reward scales and strong critic divergence sensitivity due to the sharpness of penalties for falling off the cliff. This pattern raises questions about whether the observed improvements generalize beyond tasks that inherently favor conservative value estimation or high-quality initialization.

- The chosen environments are small and largely deterministic. As a proof of concept, the inclusion of smaller environments makes sense, but the lack of consistent performance improvement over simpler purely online methods in these small test cases alongside the absence of larger-scale or higher-dimensional tasks limits the contribution of this work.

- LORO discards the LLM-collected dataset after pretraining and uses standard SAC or Double DQN for subsequent online learning. This design lacks any mechanism to preserve or regularize around the pretrained policy, making the approach vulnerable to catastrophic forgetting once new online data is introduced. In addition, vanilla SAC is known to suffer from overestimation when trained offline without corrective regularization or online interaction [1]. The paper addresses the issue of overfitting, but does not explain how this phase remains stable or avoids value overestimation given that the pretraining stage involves no environment interaction.

[1] Hussing, Marcel, et al. "Dissecting deep rl with high update ratios: Combatting value divergence." arXiv preprint arXiv:2403.05996 (2024).

**Questions:**

How do you prevent catastrophic forgetting of the pretrained policy once online training begins and the LLM data is discarded?
Given that standard SAC is prone to severe Q-value overestimation under offline training, what steps (if any) were taken to ensure stable pretraining?
Given the mixed performance of your method why did you not test across a broader suite of tasks or were you limited by the number of wrappers available for Gym environments?

---

> ### Author Response · Authors · 2025-11-25
> **Response to Reviewer ZgGw**
>
> We thank the reviewer for highlighting the reproducibility and code quality of our submission. We address your specific questions below:
>
> 1. Unpolished manuscript: We thank the reviewer for the feedback. We will ensure this is addressed in future versions.
>
> 2. Reliance on Wrappers, manual effort spent on prompt design: We agree that text wrappers are a dependency.
> However, the field is moving rapidly; thus, manual text wrapping will likely require less effort, perhaps using another LLM to help design the text wrapper. Still, we think that the core logic of LORO (using the LLM to generate a warm-start dataset for RL) will remain valid.
>
> 3. Susceptible to reward hacking and prompt engineering failures: We did notice a performance drop when the multiple-choice action contains "0", which heavily biased the choice of the LLM, as known in the literature. We hope that, as the field progresses, this will be less of an issue. Still, this is a side discussion and beyond the scope of this paper.
>
> 4. Assumption 1 lacks a strong theoretical or mechanistic justification: Verifying this assumption is one of the key contributions of this paper. We address this question in the global answer. Overall, we don't have a clear theory of *why* LLM has good coverage. We make a hypothesis based on observation and gather empirical data to further verify the hypothesis. More empirical evidence on high-dimensional, continuous action environments and theoretical explanations are left for future work.
>
>
> 5. Regarding the CliffWalking coverage measurement: The coverage coefficient here is measured based on the theoretical analysis in [Song et al, 2022]. We agree that the current evaluation protocol can be improved. Currently, we cater to the cases where the optimal policy trajectory distribution is multimodal (FrozenLake in Figure 4), thus making the measurement in CliffWalking imprecise. This is why we show the traces in Figure 4 and Figure 8 to visually justify the coverage assumption. We will update the coverage calculation in future versions.
>
> 6. Statistical significance of LORO: We address this question in the global answer.
>
> 3. Catastrophic Forgetting and Overestimation with SAC: We agree with your assessment. This is about algorithm design, while important, we want to once again highlight that the focus of our paper is about how LLM-collected data can help boost sample efficiency in RL. The LORO framework can combine with any RL algorithm, which we demonstrated with both the vanilla pretraining and AWAC in section 5.5.
>
> We hope these answers satisfy your technical questions, and we hope you consider raising your score.

---

### Official Review · Reviewer_BdGa · 2025-10-30

**Soundness:** 3
**Presentation:** 3
**Contribution:** 2
**Rating:** 2
**Confidence:** 4

**Summary:**

The paper proposes LORO: LLM Offline, RL online, a new algorithm for RL by using LLMs to bootstrap the initial policies.
LORO works by using the LLM and the problem description to generate an offline dataset. There is one hyperparameter \tau that controls the total episodes to run to collect the dataset.

This offline dataset can then be used by any RL algorithm to bootstrap the initial policy. Once bootstrapped, the RL algorithm can proceed as normally while adding new samples to the dataset.

The authors conduct an empirical evaluation on LORO to showcase its effectiveness across different discrete adn continuous environments and and also with zero-shot LLM policies to showcase the efficacy of LORO over coldstart RL methods and LLM policies.

Finally, the authors provide some light theory on domains where LLMs could perform well and not although this is not empirically demonstrated.

**Strengths:**

1. The paper is clear and well written. Enough information and contrast with related work is there.

2. The idea is intuitive and clear

3. I liked the theory part where the authors hypothesize where LLMs could do well.

4. Some results on the ablations are interesting and surprising.

**Weaknesses:**

I think this paper is almost there but not quite ready yet for publication.

1. The idea is good and the choice of baseline environments is good, but the RL algorithms themselves are lacking. Firstly, why use SAC instead of more SOTA RL methods as baselines for online RL.

2. A minor weakness: There are no experiments with domains that already have offline datasets. It would be interesting to see results here.

3. The hypothesis is good for where LLMs could do better but domains where LLMs cannot do well is not really explored so this contribution is quite limited.

4. Table 5 shows quite a heavy compute investment to generate the dataset. For example, Qwen-7b took 20 hrs for Cartpole. DQN takes maybe 15 min to learn a very good policy for CartPole that can generate an "offline" dataset for another RL algorithm. Why would one use LORO when they could use another algorithm to generate an equally good if not better dataset in a fraction of the time (assuming the simulator is cheaper to execute for 20 hrs than GPUs which at least is the case for the environments you tried).

5. Overall results are not super impressive. Looking at Table 5, the avg rank is same whether using online RL or LORO. I think your idea has merit but perhaps a better empirical design might make it shine better.

6. Bold claim where it is stated that LLM's model size and performance is not having any clear link. I think this claim is clearly overstated and might be incorrect. Ideally, you show a plot of LORO performance with many different model sizes including the tiniest LLM (1b models or even smaller). In the context of the paper you might be right but that is only with 2 models you tested. The only interesting contribution here is that the inflection point is likely 7B beyond which you will not find meaningful gains although this does not need as much text in the paper as it is obvious from the results. This would be a better contribution had you tested more models I think.

**Questions:**

Please addresses my weaknesses. Overall interesting work. Happy to engage in discussion and increase my score. (#4 is my biggest concern and primary reason for a score of 2)

---

> ### Author Response · Authors · 2025-11-25
> **Response to Reviewer BdGa**
>
> We appreciate the reviewer's positive comments on the clarity and intuition of our work. We address the weaknesses you identified below:
>
> 1. Choice of algorithm: We addressed this question in the global answer. In short, our LORO framework can be combined with any base RL algorithm. We welcome the reviewers to suggest other, more recent SoTA-based RL algorithms, and we are happy to compare LORO with them.
>
> 2. Experiment on domains with existing offline datasets: We focus on using LLM to collect data to warm start online RL, where no pre-existing offline dataset is available.
>
> 3. What happens when LLMs do not do well? We observe that when directly using LLM to make decisions, it usually does not do well (dashed lines in Fig. 3); the data collected by LLM is still useful in warm-starting RL. Identifying applications where the data collected by LLMs is not useful is an interesting future direction.
>
> 4. Compute Investment: We agree that for CartPole, using LLM to collect data may be overkill. But in more complex tasks where simulators are lacking, and each action taken has real-world consequences (e.g., medical treatment, self-driving cars), improving the sample efficiency of RL is important.
>
> 5. The ranking result is not impressive: We addressed this in the global answer. In short, the ranking doesn't show a clear picture, which can be seen in Figure 3, where LORO clearly outperforms other baselines for 4 out of 6 environments. We will remove the ranking table in the final version to reduce confusion.
>
> 6. Model Size causes no improvement claim:
> We tested the 7B and 32B size models (and the DeepSeek 14B model in one of our experiments as well). It's known in the literature that models smaller than 7B behave differently ("inflection point" as mentioned by the reviewer). It's also known that 7B to 32B is a big jump, which shows consistent performance improvements in a wide range of NLP tasks.
> In our experiment, besides the increase in size, we tested different reasoning strategies as well (SFT, CoT, Long CoT). Doing all these experiments on six different environments shows no statistically significant changes compared to the vanilla LLM with CoT.
>
> We hope this explanation clarifies your questions, and we hope you may consider raising your score.

---

> > ### Comment · Reviewer_BdGa · 2025-11-25
> >
> > 1. I think popular algorithms like PPO etc and from open-source implementations from stable baselines or other OSS packages would be essential.
> >
> > 2. I dont think this is really the case though right? How can you guarantee that no offline dataset is available for Cartpole etc that are well-studied problems which the LLM is already trained on.
> >
> > 4. I think the total time to collect the dataset is a bit too much for your current empirical settings. I agree with you that in more complex domains it might be different but then the onus is on you to showcase such domains. As it stands in the current paper, one could argue that using other online RL algorithms is the way to go for generating these datasets since the time/compute required to collect datasets with LLMs is too much.
> >
> > 5. I dont see how LORO outperforms other baselines in 4/6 domains. Most domains have the same aggregate reward. I think the plots are too jagged too. Smoothening them would help see a clear picture. I dont recommend removing the ranking table without justification. Please justify why that table and the figure have a discrepancy in interpretation then.
> >
> > 6. I think it is important to clarify that it is for these environments in the text then. The paper's text makes it seem like a general result applicable to other environments too i think.
> >
> > I'd like to see results for #1 and a better explanation for #4 and #5 before I can increase my score.

---

> ### Author Response · Authors · 2025-11-25
> **Thank you for your feedback!**
>
> 1. As stated in our reply, our LORO framework can be combined with any base (off-policy) RL algorithm. Besides the simple pretrain-fine-tune approach reported in Figure 3, we also apply the LORO framework for the popular AWAC algorithm to show that utilizing LLMs collected data can improve the learning efficiency. This is reported in Section 5.5 and Figure 7, 9, 10.
>
> 1.5 About PPO: Our LORO framework can be applied to any base off-policy RL algorithm. This is required because LORO warm-starts from the data collected by the LLM policy (i.e. transferring the learned Q function).
>
> 2. Our aim is to use LLMs to boost learning efficiency in problems where no offline datasets exists. Due to limited computation budget, we demonstrate this by experiment on the six simple environments in the paper. Although there are available offline datasets for these, as well as the simulation cost is negligible, our experiment demonstrate the potential to generalize to problem without offline datasets.
>
> 3. Similar to the previous answer, we don't claim to be the best algorithm to solve these six environments. Our goal is to demonstrate that we can utilize LLMs to collect high-coverage data and increase the sample efficiency of any base RL algorithm. We empirically demonstrate this by testing two basic algorithms (pretrain-fine tune + AWAC) on six simple environments.
>
> You can think of our work as a Proof-of-Concept (PoC), proposing a novel direction of utilizing LLM (by collecting a high coverage dataset). This PoC is demonstrated in simple environments to show the potential to generalize to more complex environments.
>
> 4. LORO outperforms the baseline (Online RL) in 4 environments (CliffWalking, Pendulum, CartPole, and MountainCar) and competitive with the other two. Since we are concerning about the sample efficiency, the right metric is the cumulative reward (i.e. the Area-Under-the-Curve of each algorithm). Figure 3 show a clear improvements over the baseline. We also reported the numeric cumulative reward of each baseline in Table 4.
>
> For a smoothed graph, please check out the Ablation Study section (E.2 to E.5). For example, Figure 11 show a smoothed curve and a clear visualization on how LORO outperforms the baseline (Online RL).
>
> 4.5 Table 1 only show the ranking, not the absolute difference between each environment. If you look at Figure 3 and Table 4, you can see that, in Pendulum and FrozenLake, LORO lose to Online RL ~1-10% of the AuC (mostly due to the warm-starting phase), while in the other four environment, LORO outperforms 30-300% Online RL in AuC.
>
> 5. While our paper empirically verified the LLM high coverage hypothesis for these six environments, we hope that this also generalizes to other more complex problems. The environments was chosen to cover both discrete and continuous action sets, both dense and sparse reward MDP. We have Pendulum that is a simplified version of robotics tasks and RepresentedPong as a simple Atari game.
>
> Once again, we thank you for your helpful feedback and we would love to continue the discussion.

---

> > ### Comment · Reviewer_BdGa · 2025-11-26
> >
> > Thanks for your response.
> >
> > Im afraid this is still a bit unconvincing.
> >
> > Firstly, please use SOTA RL algorithms from the literature. I mean DDPG, PPO etc or other more recent works for these envs.
> > Using SAC as a single online RL baseline is not enough i think.
> >
> > Regarding 4.5
> > I dont agree with this. The plots are misleading since they are looking at two very different compute units. On one hand, yes, the sample efficiency is good after the warm-start phase around 10 episodes on the x-axis.
> >
> > However, this is assuming running one episode on the simulator using online RL is the same compute cost as 1 episode of warm-start data generated by LORO. Clearly, this is currently not the case. There is no amortization analysis provided and thus currently, it does not seem efficient to use LORO in such environments (my previoius comments that it takes too much compute to generate the dataset etc)
> >
> > I do agree with your overall justification that large foundational models can generate better data and might work in more complex environments. As a result, Im afraid that your current empirical design on toy domains does not validate this hypothesis. You would need to show a domain where online RL fares far worse w.r.t. the compute needed as the x-axis too i believe. Amortizing the compute is essential to ensure that online RL is not penalized in the analysis (and i do not mean the pre-training cost of LLMs but rather the cost to use a pretrained LLM to generate a dataset).
> >
> > Happy to discuss this further.

---

> > > ### Author Response · Authors · 2025-11-29
> > > **Thank you for continue with the discussion**
> > >
> > > We want to thank the reviewer for recognize that "the sample efficiency is good after the warm-start phase".
> > > Addressing your concern:
> > >
> > > 1. DDPG baseline: We have run a quick test on the ddpg base algorithm. The figure can be seen [here](https://drive.google.com/file/d/1jCmx8t6jj7VO9fPrmH7yHknhOrpI29H9/view?usp=sharing).
> > >
> > > The cumulative reward is LORO: -194318, Online RL: -178404, Mix data w/o pretrain: -169781. Even though LORO doesn't have the highest cumulative reward, mixing the LLM collected high-coverage data in the replay buffer still boost the performance, showing the benefit of high-coverage data. We want to reiterate that our novelty is *not* the algorithm design, but the benefit of utilizing the LLM collected data to increase sample efficiency in RL.
> > >
> > > 2. Computationally expensive: from our understanding, the reviewer is concerned about the (inference) cost of using LLM to collect high-coverage data. We agree that it's overkill for these simple environments. Still, we want to highlight that our paper focus on the Sample efficiency in RL (check title). To the best of our knowledge, we are the first to propose a method to trade computation with sample efficiency by using LLM to collect high-coverage data, which has a lot of practical application when the data collection cost is much higher than LLM inference cost (e.g. medical, robotics, self-driving). As we don't focus on the computational cost, all of our experiment x-axis visualize the number of sample required to achieve the y-axis performance. Investigating how to increase sample efficiency with a smaller computational cost is an interesting question, which we left for future work.
> > >
> > > Once again, we thank the reviewer for your discussion. We hope that our answer satisfy your inquiry and happy to discuss more. We also hope that you will consider the contribution of our paper (a novel method to utilize LLM to boost sample efficiency and show that LLM collected data has high-coverage) and raise your score.

---

### Official Review · Reviewer_8NaS · 2025-11-01

**Soundness:** 2
**Presentation:** 1
**Contribution:** 1
**Rating:** 0
**Confidence:** 5

**Summary:**

This paper introduces a method to leverage priors from LLMs to generate datasets of experience for RL agents. The method is extremely simple: directly prompt an LLM to generate actions, actually take them in the environment, and then use this generated dataset to initialize an off-the-shelf RL algorithm.

**Strengths:**

- Leveraging prior knowledge from LLMs to make RL more sample efficient is a very promising research direction.
- The method proposed is extremely straightforward.
- Code to reproduce experiments is made available.

**Weaknesses:**

- The paper only shows results on toy environments, and doesn’t discuss how it would be possible to extend the method to more complex tasks.
- The paper ignores all recent literature on applying RL directly to LLMs. For domains in which the LLM already gives some reasonable actions, why not directly improve it as a policy?
- The writing in the paper suggests that the intention is to be in the offline-to-online RL regime. However, the LLM actually takes actions in the **real** environment in order to collect the “offline” dataset. This is certainly not a bad thing by itself, but it is misleading to call this offline learning, since there is interaction with the environment. The x-axis of every plot should be adjusted to account for these episodes of interaction.
- For 4 out of 6 tested environments, LORO does not perform differently than “Online RL” to a statistically significant extent.

**Questions:**

What adaptations would be needed to make LORO work for more realistic tasks?

---

> ### Author Response · Authors · 2025-11-25
> **Response to Reviewer 8NaS**
>
> We thank the reviewer for the assessment. We address your concerns below:
>
> 1. Simplistic environment: We address this question in the global answer.
>
> 2. Literature on applying RL directly to LLMs: We have discussed the direction of finetuning LLMs in the Extended Related Work under the paragraph 'Using LLM to provide extra information for RL' section (Line 698). In addition, our setting focuses on sample efficiency; thus, we don't have enough data to directly fine-tune the LLM (or a transformer-based policy in general). We demonstrated this by comparing with the SFT baseline in Line 427 and Section E.5 in Line 1456. These experiments have a budget of 10 episodes -- we expect that if we use RL finetuning (e.g., REINFORCE or PPO) on a larger amount of data, the running time would be about hundreds of GPU hours, which significantly exceeds our computational budget.
>
> 3. "Offline" Terminology: We agree with the reviewer. We use this term to connect to the Hybrid (Offline-to-Online) RL discussion, where the coverage was studied previously (Song et al, 2022). We will change the name to ``LLM data collection to warm-start online RL'' in the future version.
>
> 4. Adjusting the x-axis of the experiments: We use a fair comparison between LORO and the other baseline: they have the same budget in the number of episodes for environment interaction (e.g., 200 episodes), as shown in the Abstract and the Preliminary section (Line 133-134). For the plot, we also explained in Lines 246-250 that the initial (10) episodes reported the LLM performance, as can be seen by the jumps in performance in Figure 3 (and other figures).
>
> 5. Statistical Significance: We addressed this question in the global answer. In short, we significantly outperform other baselines in 4 out of 6 environments.
>
> 6. Extended for more complex tasks: we address this question in the global answer.
>
> We hope these clarifications address your concerns regarding the positioning of our work, and we hope you consider raising your score.

---

### Official Review · Reviewer_PrjW · 2025-11-01

**Soundness:** 1
**Presentation:** 2
**Contribution:** 1
**Rating:** 2
**Confidence:** 5

**Summary:**

This paper suggests using an LLM to collect data to warm-start an RL algorithm. The method is called LORO and is shown to have some advantages. The paper is lacking in many aspects:

1. There is not much novelty in this paper. The idea of collecting data using LLMs has been a standard application. It might be that its use in RL - to collect an offline dataset - offers some advantages, but I don't see this as a novel idea per se. Also, my decision in rejecting this paper is not based on that in general.

2. The authors seem to miss a complete literature of model-based RL and Bayesian RL that work very well in the environment they have suggested. For example, in CartPole, PILCO (https://mlg.eng.cam.ac.uk/pub/pdf/DeiRas11.pdf) can solve the task of swinging up and balancing in 3 episodes. Of course, there are many other model-based RL algorithms that need to be cited and compared to if we are discussing sample efficiency. The same goes for Bayesian RL - the idea of using a prior policy to learn has been studied. Not in the way the author suggests, i.e., collect offline data, but more to do with variational inference/posterior sampling to improve sample efficiency. The authors need to cite and compare to those for me to consider this paper. Additionally, there has been work on using LLM priors for RL, which is cited in the paper; e.g. https://arxiv.org/abs/2410.07927. The suggested method's improvement in sample efficiency should consider an empirical comparison. I really think many baselines are missing.

3. Assumption 1 is very heavy: I mean, if the LLM can cover the optimal policy, then you are more than halfway there to solve your problem. That is why I prefer the notion of a prior, whose samples get weighted by Q-values or rewards. Maybe the authors can help me understand why assumption 1 is a good assumption and why it actually holds in the real world.

4. The experiments are rather weak and simplistic. I'd suggest that the authors run some large problems to see the effect of their proposed method. Something around reasoning would be great. Maybe, even consider ALFWorld. Additionally, I would urge the authors to conduct statistical significance tests. I am not sure if the results presented are statistically significant, especially since LORO doesn't always win.

5. Are the rewards considered in the experiments sparse or dense? I didn't fully understand this point. If they are dense, the authors should consider running on sparse reward setups, since this is where the advantages might lie.

6. The authors note that when Assumption 1 is violated, LORO “typically” remains robust—yet the experiments are still in small domains. I would like to really see any scalable application of this method.

I don't know if it is my browser, but the PDF dimensions seem not right.

**Strengths:**

See above

**Weaknesses:**

See above

**Questions:**

See above

---

> ### Author Response · Authors · 2025-11-25
> **Response to Reviewer PrjW**
>
> We thank the reviewer for their detailed feedback. We address your specific concerns below:
>
> 1. Novelty of LORO and Assumption 1: See our global response. In short, our focus is on verifying the utility of the LLM collected data, which is the key novelty of this paper.
>
> 2. Simplistic environment: We also address this question in the global answer.
>
> 3. Reward type: The environments cover different types of RL problems. Pendulum gives dense reward signals. FrozenLake and CliffWalking give sparse reward when reaching the goal, with the living cost -1 for CliffWalking and 0 for FrozenLake.
>
> We hope this clarifies our contributions, and we hope you may consider raising your score.

---

> ### Comment · Reviewer_PrjW · 2025-11-25
> **Thank you**
>
> I thank the authors for their time in drafting the response. However, I don't think this answered my concerns.
>
> 1. More experiments on more realistic environments are needed. I am not convinced of the setting the authors considered. I asked for some comparisons to model-based RL; I didn't find those in the response. Might I have missed it?
>
> 2. There is some speculation about the benefits of LORO in the response, e.g., LORO can be useful in scenarios where environment interaction ... If you could show me so, I would have considered raising my score.
>
> As it stands, I maintain my current score of reject for this paper.

---

> ### Author Response · Authors · 2025-11-26
> **Thank you for your feedback**
>
> Thank you for continuing with the discussion
> 1. As we addressed in the global response and the official comment, our focus is on verifying the utility of the LLM collected data, which is the key novelty of this paper. Our LORO framework can be combined with *any* base (off-policy) RL algorithm. Besides the simple pretrain -- fine-tune approach reported in Figure 3, we also apply the LORO framework for the popular AWAC algorithm to show that utilizing LLMs collected data can improve the learning efficiency. This is reported in Section 5.5 and Figure 7, 9, 10.
>
> Since our focus is on the high-coverage data used for warm-starting and not the algorithm design, we think that our experiments have sufficiently demonstrate this hypothesis.
>
> 1.5 Model-based RL Our LORO framework can be applied to any base off-policy RL algorithm. This is required because LORO warm-starts from the data collected by the LLM policy (i.e. transferring the learned Q function). In theory, instead of learning an initial Q function, we can learn an initial transition dynamic and reward model using the LLM collected data and transfer it to continue fine-tuning in the online phase. Although it is out of scope for the current paper, showing the LORO framework is algorithm agnostic and can be extended to more algorithms besides what we show in Section 5.5 and Figure 7, 9, 10 is an interesting direction. We would love to hear your suggestion on which simple baseline would be appropriate.
>
>
> 2. The speculation mentioned is about the capability to generalize to more complex problems. For simple environment like CartPole, using LLM to collect data may be overkill. But in more complex tasks where simulators are lacking, and each action taken has real-world consequences (e.g., medical treatment, self-driving cars), improving the sample efficiency of RL is important. Our experiments have shown that we can trade computation for sample efficiency. Furthermore, to show the potential to generalize to more complex tasks, the environments was chosen to cover both discrete and continuous action sets, both dense and sparse reward MDP. We have Pendulum that is a simplified version of robotics tasks and RepresentedPong as a simple Atari game.
>
> 3 More realistic environment: As addressed in the global response, we have limited computation budget. It takes 70 hours on an H100 GPU to run the Pong experiment, and the policy hasn't even converged yet. We will investigate the utility of warm-starting RL in more complex environments, such as ALFWorld or MuJoCo, in future work. As mentioned above, we have covered a wide range of environments to show the potential to generalize to more complex task. If you have any suggestion on which simple environment that is interesting and useful, we would love to test it as well.
>
> Once again, we thank you for your helpful feedback and we would love to continue the discussion. We hope consider raising your score as well.

---

### Author Response · Authors · 2025-11-25
**Global Response to All Reviewers**

We thank the reviewers for their detailed feedback and constructive criticism. We are encouraged that the reviewers recognized the novelty and promise of the core idea (Reviewer 8NaS, ZgGw), the clarity of the writing and intuition (Reviewer BdGa), and the reproducibility of the work (Reviewer ZgGw).

- A common question across reviews is about the novelty of the paper. We want to highlight that our contribution is to verify whether LLMs can provide a dataset with good coverage to boost RL data efficiency. This research question is mentioned clearly in Lines 70-72. In short, we conducted extensive experiments to verify Assumption 1. Hence, our contribution is in raising and verifying the LLM coverage hypothesis, *not* the LORO algorithm. LORO is a general framework that can be applied to *any* RL learning algorithm (which we verified with the simplest pretrain-finetune and AWAC (Section 5.5 Line 416-424) approaches). Since our focus is on the quality of data used to warm-start RL, the baselines we used are data collected by taking actions uniformly at random and the data collected by Online RL algorithms. There are many other algorithms that can benefit from higher-quality warm-starting data, including Model-Based RL (PILCO) or Bayesian RL, as rightfully mentioned by Reviewer PrjW. As we demonstrated that our approach is robust to the choice of base RL learners in Section 5.5, extending the experiments to include these experiments is left for future versions.

- In addition, the reviewers also questioned the statistical significance of our approach. As seen in Figure 3, LORO performs roughly on par with Online RL for 2 environments and significantly exceeds Online RL and other baselines for 4 out of 6 environments (CliffWalking, CartPole, MountainCar, and RepresentedPong). Furthermore, in all experiments, we plotted and reported the results with one standard error over five random seeds (Line 256).



- Multiple reviewers also asked about the practicality of Assumption 1 and how we can extend this work for more complex problems. Verifying Assumption 1 is one of the key contributions of this paper. This may not be a strong assumption since the LLM policy does not need to cover the exact optimal policy, but some reasonable policy with constant suboptimality. In addition, since our approach of warm-starting with high-coverage data is algorithm agnostic, as demonstrated in Section 5.5 (Line 416), this approach can be applied to any base RL algorithm in other complex settings.

- We also received comments on the simplicity of the experiment environments. As Reviewer ZgGw said "our experiments are carefully designed" and BdGa "the choice of baseline environments is good". Our environment choice is closer to standard RL problems than specific text-based reasoning tasks like ALFWorld, where LLMs excel. Furthermore, our computation budget is limited. It takes 70 hours on an H100 GPU to run the Pong experiment, and the policy hasn't even converged yet. We will investigate the utility of warm-starting RL in more complex environments, such as ALFWorld or MuJoCo, in future work.

- Another common question was the comparison of computational cost versus sample efficiency. We want to clarify that, although we experimented on toy environments, LORO can be useful in scenarios where environment interaction is expensive (e.g., real-world robotics, medical treatment), but computational resources (GPU time for LLM inference/training) are relatively cheap. While LORO requires more GPU compute to generate the initial dataset than a random policy, it significantly reduces the number of steps of environment interaction required to reach optimal performance, which is the primary bottleneck in the targeted domains.

---

### Meta-Review · Area_Chair_o6Xv · 2026-01-05

**Summary:**

The paper studies warm-starting reinforcement learning using collected data by a large language model (LLM). To this effect, it presents an algorithm called LORO, which works by using the LLM and the task description to generate dataset. In doing so, the paper hinges on the assumption that the suggested policy by LLM has sufficient coverage of an optimal policy. The performance of LORO is assessed via empirical evaluation on some small environments to provide proof-of-concept.

The reviewers appreciated the overall idea behind the presented algorithm and its underlying intuition and promise, which renders relevant in RL applications. They also admired the reproducibility of the empirical assessment and code availability. However, they raised some critical concerns such as: (i) evaluations are done on toy environments and even then there's no consistent performance improvement over simple methods; (ii) reliance on wrappers and external frameworks that are not yet fully there; (iii) limited verification of the key coverage assumption (Assumption 1) in complex domains; and (iv) some missing baselines.

While the rebuttal addressed some of these, they issues still remain, in my opinion. Overall, we collectively think that the core idea is promising, novel, intuitive, and sound, but in terms of evaluations and overall execution, the paper is not yet there. Therefore, we decided to recommend rejection.

**Reviewer Concerns:**

Some reviewers were concerned that the experiments were done on toy environments, and the paper lacks discussion on how the method extends to more complex tasks. The rebuttal claimed that the goal here is to provide a proof-of-concept. Regarding this, Reviewer ZgGw, thinks (and I agree) that even though this makes sense for a proof of concept, there's evident lack of consistent performance improvement over simpler purely online methods. This concern still stands. Further, as the authors verify, the chosen environments are closer to standard RL that text-based reasoning tasks. Overall, the reviewers were eager to see evaluations on more complex domains.

Another concern was reliance on wrappers (Reviewer ZgGw), namely, the approach relies on external framework that introduces a strong dependency on environment-specific engineering. The authors agreed, arguing that due to a rapidly moving field, this limitation will be less of an issue in the future. I therefore think that this concern is not fully addressed.

Also, there were concerns regarding Assumption 1 (the coverage assumption), especially for complex domains. The authors confirm that they lack a clear theoretical reasoning, leaving more empirical evidence on high-dimensional, continuous action environments for future work.

Some concerns were regarding missing baselines in the empirical evaluation (e.g., Reviewer BdGa) and questions about statistical significance. In particular, it was mentioned that SOTA (such as DDPG and PPO) must have been used in place of SAC. This concern was mostly addressed.

**Reviewer Scores:**

- Reviewer PrjW: I believe they would maintain their score.
- Reviewer 8NaS: I believe they would maintain their score, although it is not unlikely that they would increase it to, at most, 2 (still, a reject).
- Reviewer BdGa: I believe they would maintain their score.
- Reviewer ZgGw: I believe they would maintain their score.

---

### Decision · Program_Chairs · 2026-01-26

Reject